## RESEARCH ARTICLE

# Housing environment bilaterally alters transcriptomic profile in the rat hippocampal CA1 region

Azusa Kubota[1], Kentaro Kojima[2], Shinnosuke Koketsu[3], Takayuki Kannon[2,4], Takehiro Sato[2,5], Kazuyoshi Hosomichi[2,6], Yoshiaki Shinohara[7,8*], Atsushi Tajima[1,2*]

**1** Department of Bioinformatics and Genomics, Graduate School of Advanced Preventive Medical Sciences, Kanazawa University, Kanazawa, Japan, **2** Department of Bioinformatics and Genomics, Graduate School of Medical Sciences, Kanazawa University, Kanazawa, Japan, **3** Department of Physical Therapy, Faculty of Medical Science, Nagoya Women's University, Nagoya, Japan, **4** Division of Computational Science, International Center for Brain Science, Fujita Health University, Toyoake, Japan, **5** Department of Human Biology and Anatomy, Graduate School of Medicine, University of the Ryukyus, Nishihara, Japan, **6** Laboratory of Computational Genomics, School of Life Science, Tokyo University of Pharmacy and Life Sciences, Hachioji, Japan, **7** Department of Anatomy and Systems Biology, Faculty of Medicine, University of Yamanashi, Chuo, Japan, **8** Department of Integrative Anatomy, Nagoya City University Graduate School of Medical Sciences, Nagoya, Japan

\* atajima@med.kanazawa-u.ac.jp (AT); yoshinohara@yamanashi.ac.jp (YS)

## Abstract

Brain asymmetry is a fundamental feature of neural organization. However, the molecular basis of hippocampal lateralization in response to environmental stimuli remains poorly understood. Here, we examined the transcriptomic profiles of the left and right hippocampal CA1 regions in rats reared under isolated or enriched housing conditions to elucidate hemisphere-specific responses and shared molecular adaptations. RNA-sequencing analysis revealed lateralized differences in the number and identity of differentially expressed genes, accompanied by distinct biological themes, as indicated by overrepresentation and gene set enrichment analysis. The left CA1 region was prominently engaged in pathways related to synaptic organization and mitochondrial function, whereas the right CA1 region exhibited enrichment in transcriptional regulation and RNA metabolic processes. Despite these asymmetries, co-expression and protein–protein interaction network analyses revealed shared molecular architectures. Immediate early genes formed consistent central hubs across both hemispheres, and a common Mecp2–Grin2b–Cdkl5–Tet3 protein interaction cluster was identified as a potential integrative regulatory module. Additional enrichment analysis of differentially expressed genes shared between hemispheres further highlighted conserved responses, particularly in synaptic plasticity and cell–cell communication. Together, these findings demonstrate that the left and right CA1 regions employ distinct yet partially convergent transcriptional programs to adapt to environmental stimuli. This coordinated molecular asymmetry provides novel insights into hippocampal lateralization and its role in experience-dependent brain plasticity.

**Data availability statement:** The RNA-seq dataset for this study has been deposited in the Gene Expression Omnibus (GEO) of the National Center for Biotechnology Information under the accession number GSE270831 (https://www.ncbi.nlm.nih.gov/geo/query/acc.cgi?acc=GSE270831). All other relevant data are within the paper and its Supporting Information files.

**Funding:** This work was partially supported by JSPS KAKEN Grant Numbers JP17H02221 and JP20H03295 (to YS), and JP23K18386 and JP23H03145 (to AT). This work was also supported by JST SPRING, Grant Number JPMJSP2135 (to AK). This work included results obtained using shared equipment under the MEXT Project for promoting public utilization of advanced research infrastructure (Program for supporting construction of core facilities) Grant Number JPMXS0440300024. There was no additional external funding received for this study.

**Competing interests:** The authors have declared that no competing interests exist.

## Introduction

Numerous animals and humans have left–right asymmetric brains; in humans and primates, the acquisition of higher brain functions is closely related to brain asymmetry [1,2]. Electroencephalography and functional magnetic resonance imaging are used to characterize time- and space-dependent functional differences between the left and right cerebrum of the human brain [3]. These functional differences are thought to be due to various responses to environmental stimuli, including task performance and stress exposure. Although lateral specialization of the brain has been extensively studied, the molecular factors and intricate mechanisms remain largely unknown.

Studies on hippocampal asymmetry suggest that the genetic mechanisms of visceral and brain lateralization are different. Chiral movement of the cilia induces a leftward nodal flow during embryonic development and consistently produces predictable asymmetry in the arrangement of the thoracic and abdominal organs in adulthood [4]. Normal ciliary function in wild-type mice is associated with an uneven hippocampus, and the disruption of ciliary motility leads to the randomization of organ laterality. In mice with arbitrarily inhibited ciliary movements, bilateral right isomerism occurs in the hippocampal circuitry regardless of the original direction of organ laterality [5]. Notably, specific gene expression patterns occasionally drive molecular variations in the brain. For example, the *iv* mouse, which has a mutation in the *Lrd* (left-right dynein) gene, exhibits disrupted organ lateralization and loss of hippocampal lateralization, ultimately resulting in spatial reference and working memory deficits [6].

In rodents, the size of the synapses of the CA1 pyramidal cells varies depending on whether they receive input from the left or right CA3 pyramidal cells [7,8]. This intrinsic asymmetry is closely associated with the differential expression patterns of glutamate receptor subunits, which, in turn, correlate with synaptic size and plasticity potential. Individual environmental experiences can further modulate this inherent asymmetry. For instance, rats reared in enriched environments characterized by running wheels, tunnels, and toys exhibit increased synaptic density, specifically in the right CA1 region, along with enhanced interhemispheric gamma coherence [9]. Environmental enrichment also leads to greater dendritic complexity, maintenance of spine density, and increased branching of hippocampal CA1 pyramidal neurons in the hippocampus [10]. These findings suggest that environmental stimulation actively facilitates functional lateralization within the hippocampal circuitry through activity-dependent synaptic and dendritic reorganization. Likewise, environmental enrichment induces profound neuroanatomical and molecular changes in other hippocampal subregions [11]. In the dentate gyrus, enrichment robustly stimulates adult neurogenesis, synaptogenesis, and survival of new neurons, while also increasing levels of neurotrophic factors and enhancing synaptic transmission. CA3, while less frequently studied, also exhibits increased synaptic plasticity and network connectivity, supporting improved pattern completion and associative memory. Furthermore, environmentally induced lateralization and plasticity are not unique to rodents. They have been observed across diverse species, including zebrafish, birds, and humans, indicating

that experience-dependent brain asymmetry is a highly conserved and functionally important phenomenon [12,13]. Therefore, to fully understand the intricate mechanisms underlying neural asymmetry formation, it is essential to consider not only the structural and electrophysiological aspects, but also the dynamic, context-dependent gene expression responses of hippocampal regions to environmental stimuli.

Differential mRNA profiles between the whole left and right rat hippocampi have been observed using microarrays [14], and interhemispheric differences in miRNA profiles have been found in the rat hippocampal CA3 region [15]. Moreover, previous reports revealed that housing environments alter gene expression in the rodent brain, including the hippocampus [16–18]. However, the genes responsible for neurological laterality in the rodent hippocampus and the environmental influence on asymmetry remain unknown. Therefore, in this study, we aimed to elucidate the transcriptomic profiles of the left and right CA1 regions and to assess their respective sensitivity to an enriched environment using RNA-sequencing (RNA-seq). Through systematic analyses of differentially expressed genes (DEGs) across hemispheric and environmental contrasts, including counts, functional enrichment, and co-expression/protein–protein interaction networks, we identified both distinct and shared molecular responses of the bilateral CA1 regions to environmental stimuli.

## Materials and methods

### Animals and microdissection

Male Long-Evans rats were reared under either isolated (ISO) or enriched (ENR) conditions. Rats in the ISO group were caged individually after weaning at 21 days of age and raised in standard cages (length, 32 cm; width, 22 cm; height, 13.5 cm). In the ENR group, five to six male littermates were housed together in a larger cage (length, 44 cm; width, 27 cm; height, 18.7 cm) with a ladder, running wheels, tunnels, and toys, the location of which was changed every 3–4 days. Hippocampal slices were prepared, as described previously, 4 weeks after rearing under the respective environmental conditions [7,8,15]. Briefly, the rats were euthanized with isoflurane, and their brains were immediately washed with ice-cold artificial cerebrospinal fluid (ACSF). After cooling in ACSF for 15 min, hippocampal slices were cut into 400-μm-thick sections using a McIlwain tissue chopper. The bilateral rat CA1 regions were manually dissected under a stereoscopic microscope (Olympus, Tokyo, Japan) using a handmade microblade. Three biological replicates were prepared for sequencing from the left and right CA1 regions. To minimize the effects of changes in the number of animals in the cage, ENR brain slices were prepared on the same day. The Animal Experimental Committee of Nagoya City University approved the study design (approval number: H29M-44). The animal study protocol complied with the ARRIVE guidelines 2.0 (https://arrive-guidelines.org).

### Library preparation and RNA-seq

Total RNA was extracted from rat CA1 region tissues using the RNeasy® Mini Kit (Qiagen, Germany), according to the manufacturer's protocol. All RNA samples exhibited an appropriate range of RNA integrity number values, as measured using a bioanalyzer (Agilent, USA). The cDNA libraries for RNA-seq were prepared from 500 ng of extracted RNA using the TruSeq® Stranded mRNA Sample Preparation Kit (Illumina, USA), according to the manufacturer's instructions. The libraries were sequenced on an Illumina HiSeq2000 to obtain 101-bp paired-end reads for each sample.

### Computational quantification of gene expression

The sequenced reads were processed using fastp [19] (version 0.23.4) to eliminate low-quality reads and trim adapter sequences. Transcript quantification was performed using Salmon [20] (version 1.10.3) with Ensembl release 113 of the mRatBN7.2 cDNA assembly. Salmon output files were converted into read-count data and summarized from transcript-level to gene-level quantification using tximport [21] (version 1.34.0). The raw gene count data for downstream analyses are provided in the S1 Data. The converted raw gene count data were normalized using tximport with the transcripts per

million (TPM) method for multivariate analysis, as described below. Hierarchical clustering was conducted to summarize sample similarities based on Pearson's correlation, and multidimensional scaling was used to visualize the relationships among the samples. TPM-normalized gene count data are provided in S2 Data.

### Differentially expressed genes (DEGs) analysis

Statistical analyses of DEGs were performed using edgeR [22–24] (version 4.4.2), which applies an empirical Bayes framework to stabilize dispersion estimates and is widely recognized as an appropriate method for RNA-seq datasets with limited biological replicates. After filtering out genes with very low expression using the edgeR::filterByexpr() function, the quasi-likelihood F-test was used to evaluate the significance of gene expression differences. An appropriate design matrix was formulated for the paired left–right sample quasi-likelihood F-test analysis. Nominal $p$-values were adjusted using false discovery rate (FDR) correction. In this step, the left group was treated as the control in the left–right comparison, and the ISO group was treated as the control in the ISO-ENR comparison. DEGs were detected using arbitrary thresholds of FDR $= 0.1$ and $\log_2$(fold-change) ($\log_2$FC) $= \pm \log_2 1.5$ ($= \pm 0.58$). Overlapping ISO-ENR DEGs between the left and right CA1 regions were visualized as Venn diagrams using the jvenn web application (http://bioinfo.genotoul.fr/jvenn) [25]. The primary assembly FASTA and genomic annotation GTF files of mRatBN7.2 were downloaded from Ensembl release 113.

### Correlation and multivariate analysis

The $\log_2$FC values of the 14,368 common Ensembl gene IDs from housing condition comparisons of the left and right CA1 regions were used for correlation analysis. To ensure the robustness of multivariate analysis, gene-wise expression variability was quantified using the mean absolute $\log_2$FC, and the top 500 genes were selected based on this metric for heatmap generation.

### Reverse transcription quantitative polymerase chain reaction (RT-qPCR)

For validation purposes, two representative DEGs identified by RNA-seq were selected for RT-qPCR analysis. The same RNA samples used for sequencing were also utilized for qPCR validation, with the analysis aimed at confirming the consistency of expression trends rather than providing independent replication. The number of biological replicates was identical to that used in the RNA-seq experiment (n = 3 per condition). Reverse transcription was performed using Rever-Tra Ace® qPCR RT Master Mix (TOYOBO, Japan). The PCR mixture was prepared with 10.0 µL of THUNDERBIRD® Next SYBR qPCR Mix (TOYOBO, Japan), 0.3 µM of forward and reverse primer, 1.0 µL of cDNA, and 7.8 µL of sterilized water. QuantStudio 6 Pro (Applied Biosystems, USA) was used for qPCR analysis. A relative quantification method [26] was used to measure the amount of each gene, with *Gapdh* as a normalizer. Details of the primer sequences used are provided in S1 Table. Welch's *t*-test was applied to account for potential heterogeneity of variance between the ISO and ENR groups, providing more reliable inference when the number of biological replicates is limited. Click-qPCR [27] was used for the statistical analysis and visualization of gene expression using the ΔCq values.

### Over-representation analysis (ORA)

ORA of ISO-ENR DEGs was conducted using g:Profiler [28] (https://biit.cs.ut.ee/gprofiler, version e113_59_p19_f6a03c19, database updated, 23/05/2025). Significant enrichment was tested across multiple categories, with Gene Ontology (GO) [29] (GO:Molecular Function [MF], GO:Biological Process [BP], GO:Cellular Component [CC]), Reactome [30], Kyoto Encyclopedia of Genes and Genomes (KEGG) [31], and TRANSFAC [32] as the transcription factor target databases and miRTarBase [33] as a miRNA motif database. Notably, driver GO terms were identified by grouping significant GO terms into relational clusters and selecting representative terms that explained the enrichment of the surrounding functions, whereas other emphasized terms were manually selected. The significance threshold was set at 0.05, following the

g:SCS method, which adjusts *p*-values for GO enrichment by accounting for the hierarchical structure of GO terms. This approach provides a more accurate multiple testing correction than standard methods, such as the Bonferroni and Benjamini–Hochberg corrections for dependent tests. This reduces redundancy and highlights key biological themes by retaining multiple leading terms for functional groups [34].

### Gene co-expression and protein–protein interaction (PPI) network analysis

DEGs identified from environmental comparisons using Ensembl gene IDs were converted to gene symbols and annotated using the g:Convert tool in g:Profiler. Next, Cytoscape [35] (version 3.10.2) and its plugins GeneMANIA [36] (application version: 3.5.3) and STRING [37] (application version: 2.2.0) were used to analyze gene co-expression and PPI networks among the converted DEGs. In the network visualization, the size and color of each node indicate the calculated betweenness centrality in the network, with larger and more intensely colored nodes representing a relatively higher centrality. The thickness of each edge indicates the normalized maximum weight score, which reflects the strength or confidence of the association between two genes or proteins. To improve clarity, isolated nodes were manually removed from the network.

### Gene set enrichment analysis (GSEA)

GSEA [38] was performed on the environmental comparison data using clusterProfiler [39–41] (version 4.14.6) and fgsea [42] (version 1.32.4), with the R package org.Rn.e.g.,db (version 3.20.0) used for genomic annotation. GO was used to characterize the biological features of the gene sets using the clusterProfiler:gseGO() function. To enhance the interpretability of the enrichment analysis, the Molecular Signatures Database (MSigDB) [43] and its R package msigdbr (version 24.1.0) were used for pathway enrichment analysis using the clusterProfiler::GSEA() function. The $\log_2$FC values calculated using edgeR were used as input, and the following parameters were specified for GSEA: minGSSize = 10, maxGSSize = 500, eps = 0, nPermSimple = 10000, pvalueCutoff = 0.1, pAdjustMethod = "BH" (Benjamini-Hochberg), and by = "fgsea." In the resulting dot plot, the GeneRatio (i.e., the ratio of core enrichment genes to the total number of genes in each gene set) was plotted on the x-axis, indicating the proportion of genes that most strongly contributed to enrichment. In GSEA, core-enriched genes are defined as the subset of genes that contribute most significantly to the enrichment score, thereby representing the central elements driving the biological changes within each gene set. These genes were then used to calculate the GeneRatio in the dot plot, indicating their contribution to primary changes in the pathway.

### Data processing, statistics, and drawing

R (version 4.4.2) and its tidyverse package (version 2.0.0) were used for data preprocessing, statistical computing, and data visualization.

## Results

### Comparison of transcriptomic similarities

Transcript quantification and summarization were successfully performed, resulting in the identification of 23,978 genes; each was annotated with a unique Ensembl gene ID. The raw gene count dataset was normalized to scaled TPM values, which were used for the analyses described in this section (S1 Data). To examine the overall transcriptomic similarity between samples, multivariate analyses were conducted based on the scaled TPM values (S2 Fig). The hierarchical clustering dendrogram revealed a clear separation between the ENR and ISO groups and the left and right CA1 region samples from the same individual clustered as pairs. Furthermore, the multidimensional scaling plot based on the scaled TPM values illustrated the relative distances among the samples.

## DEG analysis and RT-qPCR

DEG analysis showed that no genes met the significance criteria in the left–right comparison under the ENR conditions, whereas only one DEG, *Pklm1*, was identified under the ISO conditions (S3 Fig). The complete edgeR results for the left–right comparison are provided in S2 Data. Conversely, the ISO–ENR comparison revealed significant differences in the gene expression profiles in both the left and right CA1 regions. In the left CA1 region, 189 DEGs were identified: 25 genes were highly expressed in the ENR group, and 164 genes were highly expressed in the ISO group (Fig 1A). In the right CA1 region, 94 DEGs were detected: 13 genes were highly expressed in the ENR group and 81 genes in the ISO group (Fig 1B). The full edgeR results for the ISO–ENR comparison are available in the S3 Data, and the corresponding DEG lists are compiled in the S4 Data.

The number of overlapping ISO–ENR DEGs between the left and right CA1 regions was visualized using Venn diagrams (Fig 2A). A total of 70 DEGs were shared, with seven upregulated in the ENR group and 63 upregulated in the ISO group. All shared DEGs are listed in Table 1. To validate the DEG analysis results, RT-qPCR experiments were carried out using the selected genes from Table 1 according to the following criteria: $\log_{10}CPM > 3$ and $FDR < 0.05$. Among the genes that met these criteria, *Cdkl5* and *Fos* genes were selected for RT-qPCR analysis. *Cdkl5* and *Fos* showed significantly lower relative expression in the ENR group than in the ISO group in both the left and right CA1 regions (Fig 2B).

To further characterize the ISO–ENR DEGs, correlation and multivariate analyses were conducted using $\log_{2}FC$ values from the edgeR analysis of the left and right CA1 regions (S4 Fig). The correlation plot of the $\log_{2}FC$ values for environmental comparisons between the left and right CA1 regions yielded a Pearson correlation coefficient of 0.587. Additionally,

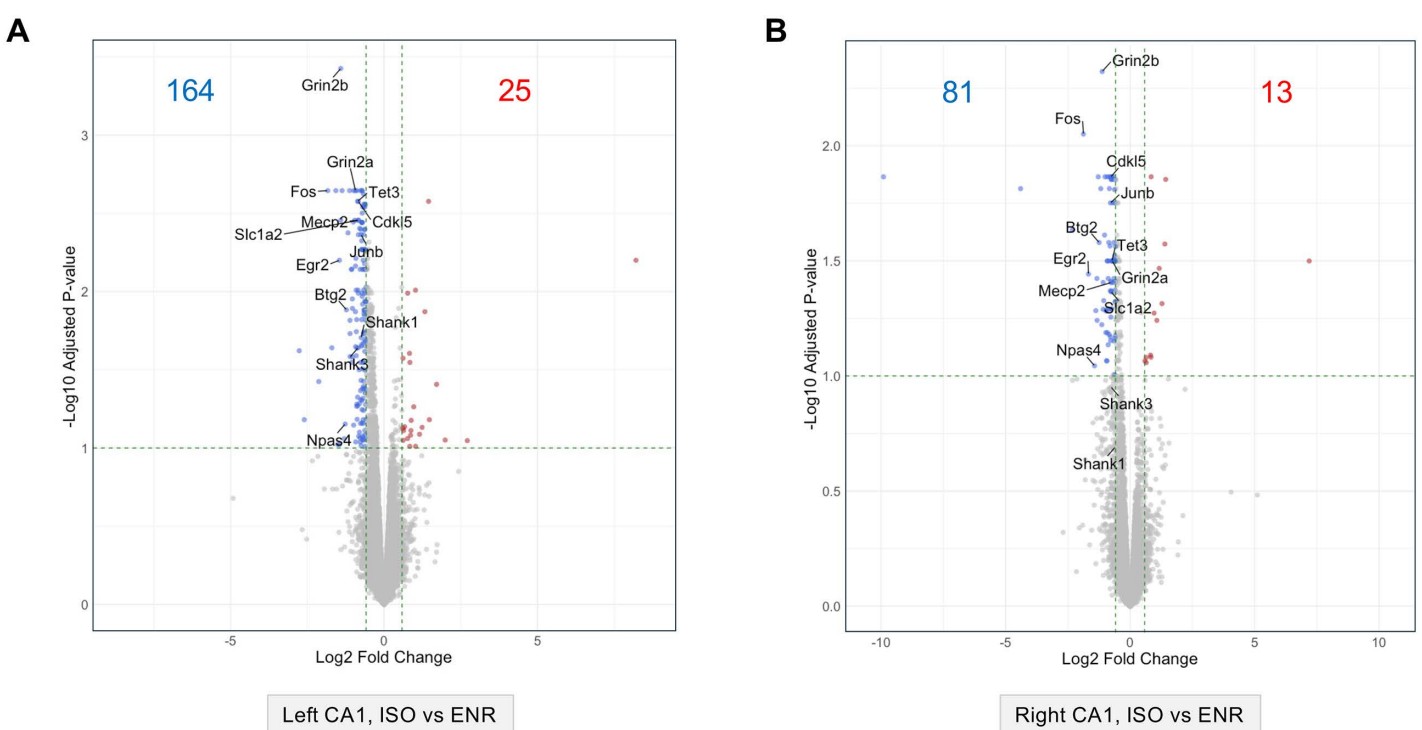

**Fig 1. Differentially expressed genes (DEGs) in the housing condition comparison between the left and right CA1 regions.** (A) and (B) show volcano plots illustrating the distribution of statistically tested genes. FDR: false discovery rate; FC: fold-change. Thresholds for DEG analysis (FDR = 0.1 and $\log_{2}FC = \pm\log_{2}1.5$) are indicated using green dashed lines. (A) ISO–ENR comparison in the left CA1 region. (B) ISO–ENR comparison in the right CA1 region. ISO, isolated housing conditions; ENR, enriched housing conditions.

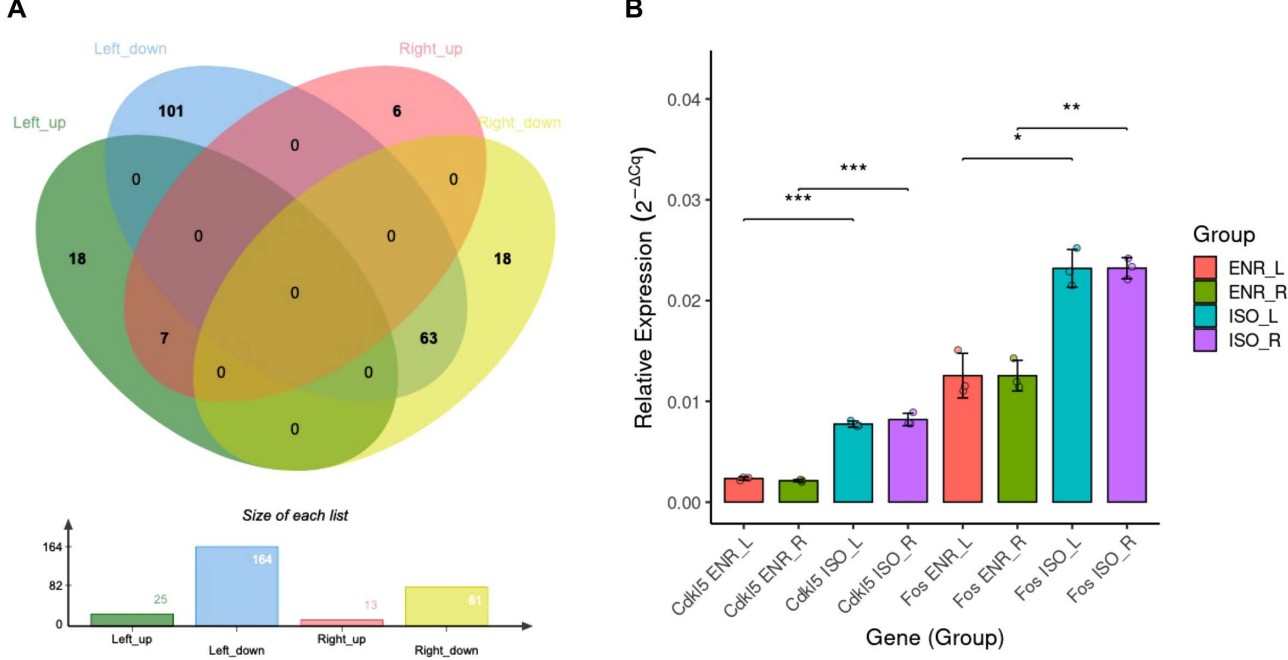

**Fig 2. Summary of bilaterally shared ISO–ENR DEGs and RT-qPCR validation.** (A) Venn diagrams showing the overlap between the ISO–ENR DEG sets in the left and right CA1 regions. (B) RT-qPCR quantification of *Fos* and *Cdkl5* expression, normalized to *Gapdh*. *$p < 0.05$, **$p < 0.01$, ***$p < 0.001$. ISO, isolated housing conditions; ENR, enriched housing conditions; DEGs, differentially expressed genes; RT-qPCR, reverse transcription-quantitative polymerase chain reaction.

a heatmap based on the top 500 genes ranked by absolute $\log_2$FC values showed a high degree of directional similarity in gene expression in the bilateral CA1 regions.

Taken together, these results indicate that although the number of environment-responsive genes differs between hemispheres (left: 189; right: 94), the overall direction of gene expression changes in response to the environment is highly correlated between the left and right CA1 regions.

## ORA of bilateral DEGs in the environmental comparison

To elucidate the biological significance of the effects of the housing environment on bilateral CA1 regions, ORA was performed using the ISO–ENR DEGs identified in the left and right hippocampal CA1 regions. All results obtained using g:Profiler are provided in the S5 Data.

Fig 3A shows the functional associations of the DEGs in the left hippocampal CA1 region as determined with GO enrichment and pathway analyses. A total of 189 DEGs were analyzed based on Ensembl gene IDs, of which 180 were annotated. This analysis revealed significant enrichment of GO-MF terms related to gene transcription and DNA binding (e.g., sequence-specific DNA binding and DNA-binding transcription factor activity), as well as membrane-associated protein functions (e.g., transmembrane signaling receptor activity, voltage-gated ion channel activity, and glutamate-gated calcium ion channel activity). Among BP terms, those associated with nervous system development (e.g., axonogenesis), synaptic function, and memory formation (e.g., long-term memory and synaptic vesicle localization) were identified. For CC, synapse- and axon-related terms (e.g., synaptic membrane, Schaffer collateral-CA1 synapse, and axon initial segment) were significantly enriched. Pathway analysis indicated significant enrichment of the MAPK signaling pathway (KEGG:04010) and the nuclear receptor transcription pathway (REAC:R-NOR-383280).

**Table 1. Shared DEGs by comparing rearing environments in bilateral CA1.**

| Gene | Description | logFC (L) | logCPM (L) | FDR (L) | logFC (R) | logCPM (R) | FDR (R) |
|---|---|---|---|---|---|---|---|
| Abca5 | ATP binding cassette subfamily A member 5 | −2.600 | 1.191 | 0.066 | −9.905 | 1.185 | 0.014 |
| Adgrf5 | adhesion G protein-coupled receptor F5 | −0.697 | 3.738 | 0.089 | −0.919 | 3.630 | 0.032 |
| Ago3 | argonaute RISC catalytic component 3 | −0.750 | 3.282 | 0.037 | −0.653 | 3.294 | 0.026 |
| Ankrd52 | ankyrin repeat domain 52 | −0.695 | 4.779 | 0.003 | −0.596 | 4.785 | 0.048 |
| Apold1 | apolipoprotein L domain containing 1 | −1.702 | 2.150 | 0.023 | −2.317 | 2.306 | 0.023 |
| Arfgef3 | ARFGEF family member 3 | −0.588 | 5.949 | 0.009 | −0.629 | 5.886 | 0.016 |
| Bach2 | BTB domain and CNC homolog 2 | −0.803 | 1.885 | 0.052 | −0.772 | 1.879 | 0.056 |
| Bmp3 | bone morphogenetic protein 3 | −0.684 | 2.255 | 0.070 | −1.332 | 2.081 | 0.038 |
| Btg2 | BTG anti-proliferation factor 2 | −1.224 | 2.917 | 0.013 | −1.244 | 2.908 | 0.026 |
| Cacna1e | calcium voltage-gated channel subunit alpha1 E | −0.891 | 7.262 | 0.002 | −0.832 | 7.217 | 0.015 |
| Caln1 | calneuron 1 | −0.728 | 4.474 | 0.005 | −0.778 | 4.363 | 0.039 |
| Cbl | Cbl proto-oncogene | −0.714 | 4.648 | 0.004 | −0.690 | 4.675 | 0.018 |
| Cchcr1 | coiled-coil alpha-helical rod protein 1 | 0.829 | 2.310 | 0.025 | 0.821 | 2.174 | 0.081 |
| Cdkl5 | cyclin-dependent kinase-like 5 | −0.840 | 5.585 | 0.003 | −0.769 | 5.549 | 0.014 |
| Cds2 | CDP-diacylglycerol synthase 2 | −0.662 | 5.235 | 0.010 | −0.737 | 5.171 | 0.014 |
| Clmn | calmin | −0.724 | 5.496 | 0.005 | −0.639 | 5.435 | 0.032 |
| Csrnp3 | cysteine and serine rich nuclear protein 3 | −0.781 | 4.124 | 0.007 | −0.602 | 4.129 | 0.099 |
| Dgkh | diacylglycerol kinase, eta | −1.414 | 3.131 | 0.003 | −0.931 | 3.096 | 0.086 |
| Egfr | epidermal growth factor receptor | −0.748 | 2.294 | 0.096 | −0.790 | 2.227 | 0.068 |
| Egr2 | early growth response 2 | −1.455 | 2.856 | 0.006 | −1.673 | 2.710 | 0.036 |
| Elfn2 | extracellular leucine-rich repeat and fibronectin type III domain containing 2 | −0.917 | 7.317 | 0.006 | −0.832 | 7.217 | 0.032 |
| Erbb4 | erb-b2 receptor tyrosine kinase 4 | −1.030 | 3.940 | 0.013 | −0.757 | 3.845 | 0.014 |
| Flt1 | Fms related receptor tyrosine kinase 1 | −0.676 | 4.561 | 0.089 | −0.882 | 4.485 | 0.073 |
| Fos | Fos proto-oncogene, AP-1 transcription factor subunit | −1.835 | 3.075 | 0.002 | −1.876 | 3.045 | 0.009 |
| Fzd3 | frizzled class receptor 3 | −0.656 | 4.324 | 0.014 | −0.597 | 4.313 | 0.018 |
| Gabrb1 | gamma-aminobutyric acid type A receptor subunit beta1 | −0.839 | 4.376 | 0.010 | −0.825 | 4.264 | 0.014 |
| Gan | gigaxonin | −1.038 | 1.889 | 0.026 | −1.137 | 1.829 | 0.060 |
| Grin2a | glutamate ionotropic receptor NMDA type subunit 2A | −0.932 | 7.760 | 0.002 | −0.717 | 7.752 | 0.032 |
| Grin2b | glutamate ionotropic receptor NMDA type subunit 2B | −1.411 | 5.835 | 0.0004 | −1.123 | 5.771 | 0.005 |
| Gucy1a2 | guanylate cyclase 1 soluble subunit alpha 2 | −1.066 | 2.955 | 0.007 | −1.007 | 3.005 | 0.014 |
| Hivep3 | HIVEP zinc finger 3 | −0.681 | 5.667 | 0.006 | −0.587 | 5.594 | 0.067 |
| Igsf9b | immunoglobulin superfamily, member 9B | −1.368 | 2.790 | 0.002 | −1.083 | 2.644 | 0.051 |
| Ino80d | INO80 complex subunit D | −1.175 | 2.272 | 0.004 | −1.025 | 2.323 | 0.024 |
| Junb | JunB proto-oncogene, AP-1 transcription factor subunit | −0.745 | 5.479 | 0.004 | −0.788 | 5.419 | 0.018 |
| Kdr | kinase insert domain receptor | −0.923 | 2.770 | 0.091 | −1.324 | 2.725 | 0.057 |
| Klf10 | KLF transcription factor 10 | −0.603 | 5.020 | 0.020 | −0.652 | 5.014 | 0.038 |
| Mecp2 | methyl CpG binding protein 2 | −0.836 | 3.695 | 0.003 | −0.764 | 3.624 | 0.039 |
| Mrtfb | myocardin related transcription factor B | −0.655 | 6.576 | 0.005 | −0.617 | 6.494 | 0.032 |
| Mtcl2 | microtubule crosslinking factor 2 | −0.953 | 4.782 | 0.003 | −0.758 | 4.723 | 0.051 |
| Npas4 | neuronal PAS domain protein 4 | −1.267 | 2.819 | 0.070 | −1.429 | 2.877 | 0.090 |

*(Continued)*

| Gene | Description | logFC (L) | logCPM (L) | FDR (L) | logFC (R) | logCPM (R) | FDR (R) |
|---|---|---|---|---|---|---|---|
| Nsd3 | nuclear receptor binding SET domain protein 3 | −0.755 | 4.227 | 0.004 | −0.753 | 4.185 | 0.014 |
| Pdzrn4 | PDZ domain containing RING finger 4 | −1.111 | 1.705 | 0.015 | −0.973 | 1.732 | 0.065 |
| Phc3 | polyhomeotic homolog 3 | −0.732 | 3.975 | 0.004 | −0.615 | 3.983 | 0.043 |
| Ppp1r9a | protein phosphatase 1, regulatory subunit 9A | −0.625 | 6.169 | 0.003 | −0.604 | 6.112 | 0.014 |
| Ptar1 | protein prenyltransferase alpha subunit repeat containing 1 | −0.931 | 2.587 | 0.007 | −0.806 | 2.737 | 0.027 |
| Ptprb | protein tyrosine phosphatase, receptor type, B | −0.695 | 4.413 | 0.084 | −0.947 | 4.258 | 0.086 |
| Rc3h1 | ring finger and CCCH-type domains 1 | −0.662 | 3.984 | 0.007 | −0.593 | 3.936 | 0.032 |
| RT1-CE15 | RT1 class I, locus CE15 | −2.765 | 1.491 | 0.024 | −4.398 | 1.570 | 0.015 |
| RT1-CE16 | RT1 class I, locus CE16 | 0.770 | 4.842 | 0.010 | 0.844 | 4.844 | 0.014 |
| RT1-N1 | RT1 class Ib, locus N1 | 1.451 | 2.212 | 0.003 | 1.430 | 2.082 | 0.014 |
| RT1-N3 | RT1 class Ib, locus N3 | 1.024 | 1.610 | 0.010 | 1.170 | 1.582 | 0.034 |
| Scn8a | sodium voltage-gated channel alpha subunit 8 | −0.720 | 7.846 | 0.002 | −0.679 | 7.784 | 0.014 |
| Slc1a2 | solute carrier family 1 member 2 | −0.836 | 9.126 | 0.003 | −0.762 | 9.074 | 0.043 |
| Slc4a8 | solute carrier family 4 member 8 | −0.834 | 4.395 | 0.004 | −0.750 | 4.283 | 0.051 |
| Slc7a14 | solute carrier family 7, member 14 | −0.623 | 4.700 | 0.004 | −0.602 | 4.645 | 0.042 |
| Ston2 | stonin 2 | −0.796 | 4.162 | 0.004 | −0.881 | 4.052 | 0.038 |
| Tanc2 | tetratricopeptide repeat, ankyrin repeat and coiled-coil containing 2 | −0.714 | 6.028 | 0.002 | −0.618 | 5.989 | 0.030 |
| Tead1 | TEA domain transcription factor 1 | −0.647 | 2.047 | 0.042 | −0.841 | 1.952 | 0.066 |
| Tet3 | tet methylcytosine dioxygenase 3 | −0.863 | 4.300 | 0.003 | −0.727 | 4.165 | 0.032 |
| Tmem245 | transmembrane protein 245 | −0.609 | 4.184 | 0.032 | −0.857 | 4.199 | 0.026 |
| Tnrc6b | trinucleotide repeat containing adaptor 6B | −1.121 | 4.268 | 0.002 | −0.918 | 4.229 | 0.014 |
| Xkr4 | XK related 4 | −0.648 | 4.784 | 0.013 | −0.668 | 4.755 | 0.014 |
| Zbtb20 | zinc finger and BTB domain containing 20 | −1.572 | 3.396 | 0.002 | −1.178 | 3.305 | 0.015 |
| Zfp704 | zinc finger protein 704 | −1.001 | 3.440 | 0.004 | −0.794 | 3.400 | 0.043 |
| ENSRNOG00000066523 | None | 1.330 | 3.232 | 0.013 | 1.397 | 3.164 | 0.027 |
| ENSRNOG00000066876 | None | −0.991 | 1.982 | 0.071 | −1.279 | 1.981 | 0.014 |
| ENSRNOG00000067970 | None | −1.034 | 2.914 | 0.011 | −0.911 | 2.874 | 0.065 |
| ENSRNOG00000069024 | None | −0.908 | 3.174 | 0.026 | −1.060 | 3.201 | 0.047 |
| ENSRNOG00000069624 | None | 1.712 | 1.187 | 0.039 | 1.280 | 0.904 | 0.049 |
| ENSRNOG00000070648 | None | 0.842 | 5.558 | 0.028 | 0.964 | 5.551 | 0.053 |

logCPM, $\log_{10}$CPM; logFC, $\log_2$(fold-change); FDR, false discovery rate. L: left, R: right; DEGs, differentially expressed genes.

Fig 3B shows the functional associations of the DEGs identified in the right hippocampal CA1 region using GO enrichment and pathway analyses. A total of 94 DEGs were analyzed based on Ensembl gene IDs, of which 87 were annotated. This analysis revealed significant enrichment of transmembrane receptor protein kinase activity and DNA methylation-related GO-MF terms (e.g., DNA 5-methylcytosine dioxygenase activity). In GO-BP, the regulation of RNA metabolic processes was highly enriched. For GO-CC, the terms related to cell junctions and postsynaptic density were significantly represented. Pathway analysis revealed significant enrichment of NPAS4-mediated transcriptional regulation.

Fig 4 illustrates the functional associations between shared DEGs identified through GO enrichment and pathway analyses. From the 70 shared Ensembl gene IDs, 64 corresponding annotated gene symbols were used in this analysis. This analysis revealed significant enrichment in GO-MF terms related to transcriptional regulator activity and transmembrane receptor protein tyrosine kinase activity. In terms of GO-BP, the most significantly enriched term was positive regulation of

**A**

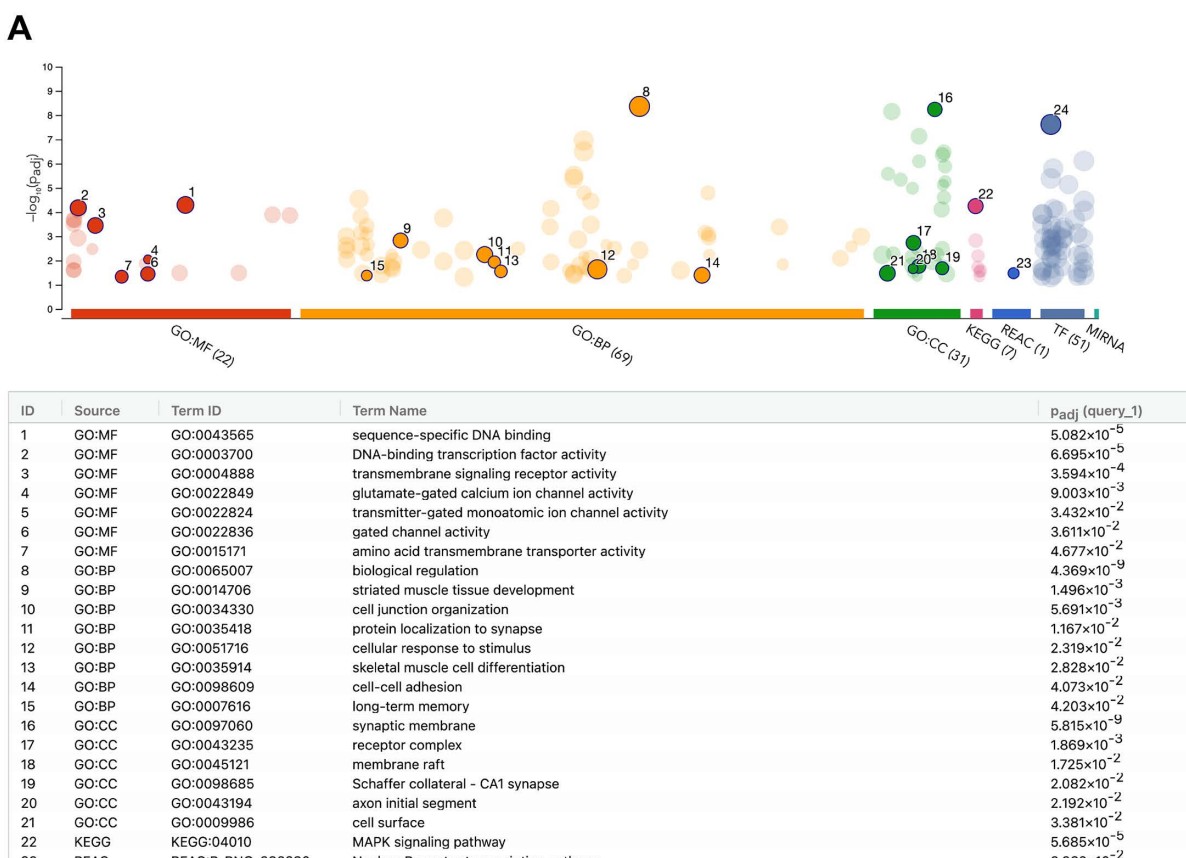

| ID | Source | Term ID | Term Name | $p_{adj}$ (query_1) |
|----|--------|---------|-----------|---------------------|
| 1 | GO:MF | GO:0043565 | sequence-specific DNA binding | $5.082 \times 10^{-5}$ |
| 2 | GO:MF | GO:0003700 | DNA-binding transcription factor activity | $6.695 \times 10^{-5}$ |
| 3 | GO:MF | GO:0004888 | transmembrane signaling receptor activity | $3.594 \times 10^{-4}$ |
| 4 | GO:MF | GO:0022849 | glutamate-gated calcium ion channel activity | $9.003 \times 10^{-3}$ |
| 5 | GO:MF | GO:0022824 | transmitter-gated monoatomic ion channel activity | $3.432 \times 10^{-2}$ |
| 6 | GO:MF | GO:0022836 | gated channel activity | $3.611 \times 10^{-2}$ |
| 7 | GO:MF | GO:0015171 | amino acid transmembrane transporter activity | $4.677 \times 10^{-2}$ |
| 8 | GO:BP | GO:0065007 | biological regulation | $4.369 \times 10^{-9}$ |
| 9 | GO:BP | GO:0014706 | striated muscle tissue development | $1.496 \times 10^{-3}$ |
| 10 | GO:BP | GO:0034330 | cell junction organization | $5.691 \times 10^{-3}$ |
| 11 | GO:BP | GO:0035418 | protein localization to synapse | $1.167 \times 10^{-2}$ |
| 12 | GO:BP | GO:0051716 | cellular response to stimulus | $2.319 \times 10^{-2}$ |
| 13 | GO:BP | GO:0035914 | skeletal muscle cell differentiation | $2.828 \times 10^{-2}$ |
| 14 | GO:BP | GO:0098609 | cell-cell adhesion | $4.073 \times 10^{-2}$ |
| 15 | GO:BP | GO:0007616 | long-term memory | $4.203 \times 10^{-2}$ |
| 16 | GO:CC | GO:0097060 | synaptic membrane | $5.815 \times 10^{-9}$ |
| 17 | GO:CC | GO:0043235 | receptor complex | $1.869 \times 10^{-3}$ |
| 18 | GO:CC | GO:0045121 | membrane raft | $1.725 \times 10^{-2}$ |
| 19 | GO:CC | GO:0098685 | Schaffer collateral - CA1 synapse | $2.082 \times 10^{-2}$ |
| 20 | GO:CC | GO:0043194 | axon initial segment | $2.192 \times 10^{-2}$ |
| 21 | GO:CC | GO:0009986 | cell surface | $3.381 \times 10^{-2}$ |
| 22 | KEGG | KEGG:04010 | MAPK signaling pathway | $5.685 \times 10^{-5}$ |
| 23 | REAC | REAC:R-RNO-383280 | Nuclear Receptor transcription pathway | $3.362 \times 10^{-2}$ |
| 24 | TF | TF:M00803_1 | Factor: E2F; motif: GGCGSG; match class: 1 | $2.416 \times 10^{-8}$ |

Left ISO_ENR: Of the 189 Ensembl gene ID-based DEGs, 180 annotated genes were used.

**B**

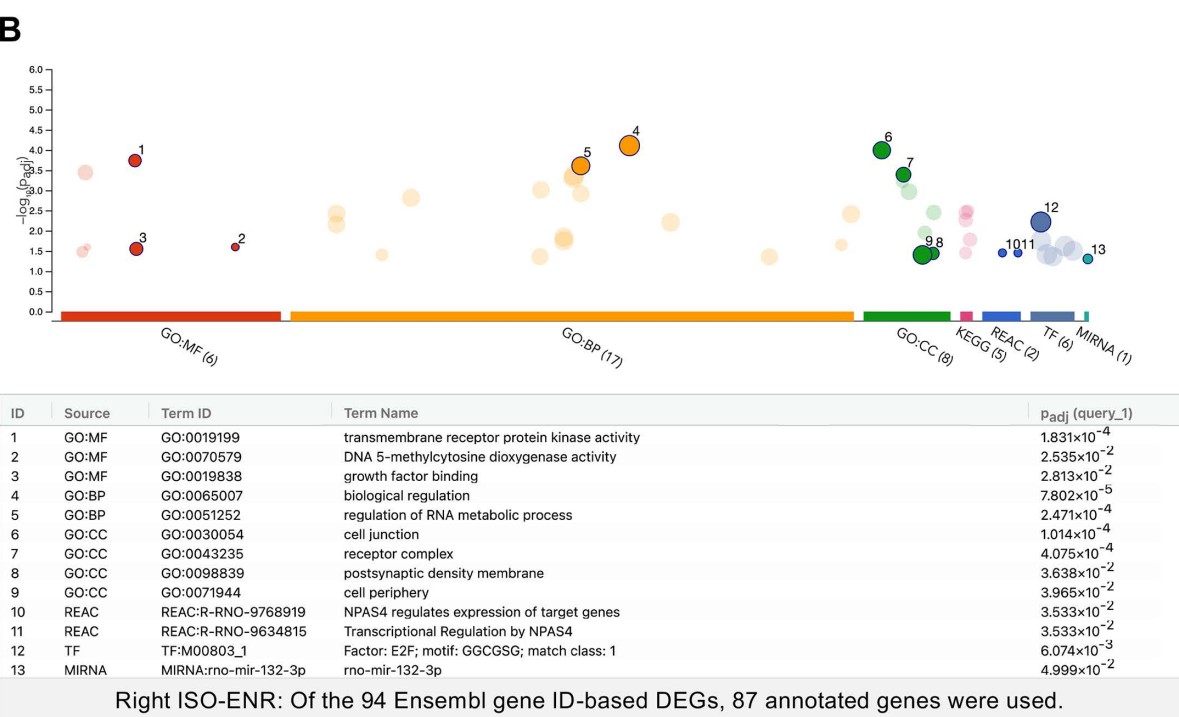

| ID | Source | Term ID | Term Name | $p_{adj}$ (query_1) |
|----|--------|---------|-----------|---------------------|
| 1 | GO:MF | GO:0019199 | transmembrane receptor protein kinase activity | $1.831 \times 10^{-4}$ |
| 2 | GO:MF | GO:0070579 | DNA 5-methylcytosine dioxygenase activity | $2.535 \times 10^{-2}$ |
| 3 | GO:MF | GO:0019838 | growth factor binding | $2.813 \times 10^{-2}$ |
| 4 | GO:BP | GO:0065007 | biological regulation | $7.802 \times 10^{-5}$ |
| 5 | GO:BP | GO:0051252 | regulation of RNA metabolic process | $2.471 \times 10^{-4}$ |
| 6 | GO:CC | GO:0030054 | cell junction | $1.014 \times 10^{-4}$ |
| 7 | GO:CC | GO:0043235 | receptor complex | $4.075 \times 10^{-4}$ |
| 8 | GO:CC | GO:0098839 | postsynaptic density membrane | $3.638 \times 10^{-2}$ |
| 9 | GO:CC | GO:0071944 | cell periphery | $3.965 \times 10^{-2}$ |
| 10 | REAC | REAC:R-RNO-9768919 | NPAS4 regulates expression of target genes | $3.533 \times 10^{-2}$ |
| 11 | REAC | REAC:R-RNO-9634815 | Transcriptional Regulation by NPAS4 | $3.533 \times 10^{-2}$ |
| 12 | TF | TF:M00803_1 | Factor: E2F; motif: GGCGSG; match class: 1 | $6.074 \times 10^{-3}$ |
| 13 | MIRNA | MIRNA:rno-mir-132-3p | rno-mir-132-3p | $4.999 \times 10^{-2}$ |

Right ISO-ENR: Of the 94 Ensembl gene ID-based DEGs, 87 annotated genes were used.

**Fig 3. Manhattan plots showing significantly enriched terms from over-representation analyses of the left and right ISO–ENR DEGs.** The numbered and emphasized points have the following characteristics: Driver terms, representing the most relevant GO terms in a tabular format, are highlighted in GO:MF, GO:BP, and GO:CC. Other highlighted points were manually selected based on their significance in additional databases (KEGG Pathway, REAC: Reactome Pathway, TF: TRANSFAC, and MIRNA: miRTarBase). The adjusted *p*-values (padj) were calculated using the G:SCS method implemented in g:Profiler. The threshold for statistical significance was set at 0.05. (A) Manhattan plot and the highlighted term details for the left ISO–ENR DEGs. (B) Right ISO-ENR Manhattan plot and the highlighted term details for the right ISO–ENR DEGs. ISO, isolated housing conditions; ENR, enriched housing conditions; DEGs, differentially expressed genes; GO, Gene Ontology; MF, Molecular function; BP, Biological process; CC, Cellular component; KEGG, Kyoto Encyclopedia of Genes and Genomes.

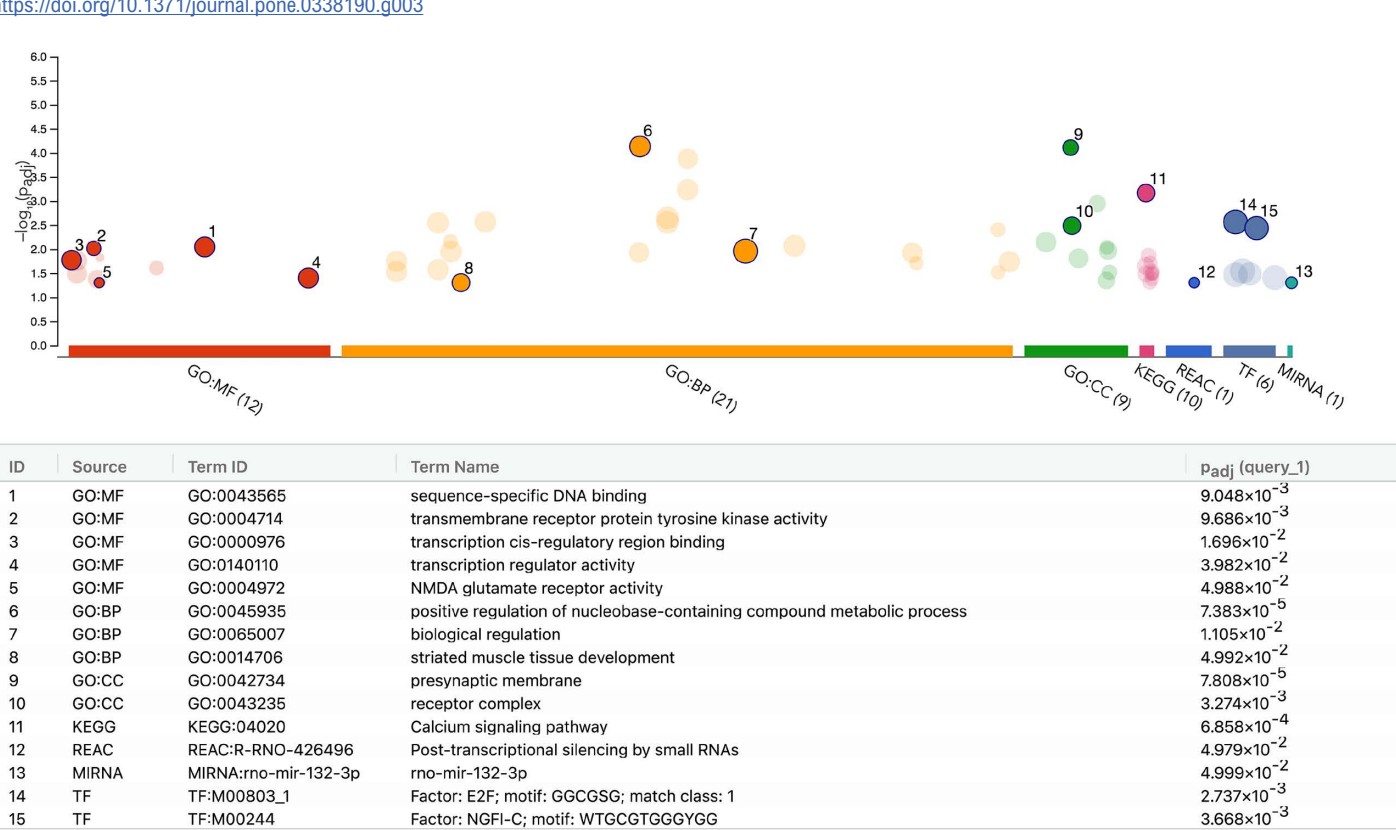

| ID | Source | Term ID | Term Name | $p_{adj}$ (query_1) |
|----|--------|---------|-----------|------------------|
| 1 | GO:MF | GO:0043565 | sequence-specific DNA binding | $9.048\times10^{-3}$ |
| 2 | GO:MF | GO:0004714 | transmembrane receptor protein tyrosine kinase activity | $9.686\times10^{-3}$ |
| 3 | GO:MF | GO:0000976 | transcription cis-regulatory region binding | $1.696\times10^{-2}$ |
| 4 | GO:MF | GO:0140110 | transcription regulator activity | $3.982\times10^{-2}$ |
| 5 | GO:MF | GO:0004972 | NMDA glutamate receptor activity | $4.988\times10^{-2}$ |
| 6 | GO:BP | GO:0045935 | positive regulation of nucleobase-containing compound metabolic process | $7.383\times10^{-5}$ |
| 7 | GO:BP | GO:0065007 | biological regulation | $1.105\times10^{-2}$ |
| 8 | GO:BP | GO:0014706 | striated muscle tissue development | $4.992\times10^{-2}$ |
| 9 | GO:CC | GO:0042734 | presynaptic membrane | $7.808\times10^{-5}$ |
| 10 | GO:CC | GO:0043235 | receptor complex | $3.274\times10^{-3}$ |
| 11 | KEGG | KEGG:04020 | Calcium signaling pathway | $6.858\times10^{-4}$ |
| 12 | REAC | REAC:R-RNO-426496 | Post-transcriptional silencing by small RNAs | $4.979\times10^{-2}$ |
| 13 | MIRNA | MIRNA:rno-mir-132-3p | rno-mir-132-3p | $4.999\times10^{-2}$ |
| 14 | TF | TF:M00803_1 | Factor: E2F; motif: GGCGSG; match class: 1 | $2.737\times10^{-3}$ |
| 15 | TF | TF:M00244 | Factor: NGFI-C; motif: WTGCGTGGGYGG | $3.668\times10^{-3}$ |

Of the 70 Ensembl gene IDs as shared DEGs, 64 annotated gene symbols were used.

**Fig 4. Manhattan plot showing significantly enriched terms from over-representation analysis of the bilaterally shared ISO-ENR DEGs.** The numbered and emphasized points have the following characteristics: Driver terms, representing the most relevant GO terms in a tabular format, are highlighted in GO:MF, GO:BP, and GO:CC. Other highlighted points were manually selected based on their significance in additional preset databases (KEGG Pathway, REAC: Reactome Pathway, TF: TRANSFAC, and MIRNA: miRTarBase). The adjusted *p*-values (padj) were calculated using the G:SCS method implemented in g:Profiler. The threshold for statistical significance was set at 0.05. ISO, isolated housing conditions; ENR, enriched housing conditions; DEGs, differentially expressed genes; GO, Gene Ontology; MF, Molecular function; BP, Biological process; CC, Cellular component; KEGG, Kyoto Encyclopedia of Genes and Genomes.

nucleobase-containing compound metabolic processes, with the smallest adjusted *p*-value. The presynaptic membrane and receptor complex were significantly represented in GO-CC. Pathway analysis revealed significant enrichment in the calcium signaling pathway (KEGG:04020) and post-transcriptional silencing by small RNAs (REAC:R-RNO-426496). Additionally, the transcription factor E2F and microRNA miR-132-3p were found to be enriched.

Collectively, these ORA results suggest that the bilateral CA1 regions respond to environmental stimuli through distinct biological processes: the left hemisphere is enriched for pathways related to synaptic function and memory formation, whereas the right hemisphere is associated with transcriptional and epigenetic regulatory mechanisms.

## Gene co-expression and PPI networks of DEGs in the environmental comparison

To evaluate the functional associations among ISO–ENR DEGs, gene co-expression network analyses were performed under three different conditions: left, right, and shared ISO–ENR DEGs. Gene co-expression networks were constructed using the GeneMANIA database in Cytoscape (Figs 5A, 6A and 7A). All results from the Cytoscape analysis are provided in S6 Data. To highlight the common features of the bilaterally shared DEGs, highly correlated genes were predicted and visualized using a query gene-based weighting algorithm (Fig 7A). Across the three co-expression networks, a cluster of immediate early genes (IEGs), such as *Arc*, *Fos*, and *Junb,* was connected by thick edges. Additionally, key hub DEGs were highlighted in each network: *Shc3*, *Tnr*, and *Taok1* in the left CA1 region; *Ppp1r9a* and *Erbb4* in the right CA1 region; *Ppp1r9a*, *Jun*, and *Fosb* in the shared DEGs.

To examine interactions at the protein level, PPI network analyses were also conducted under the same three conditions. PPI networks were generated using the STRING database in Cytoscape (Figs 5B, 6B, and 7B). In these networks, Fos consistently appeared as a central hub protein. In addition to the IEG protein cluster, PPI network analysis revealed strong connections that were not observed in the co-expression networks, including Fos–Npas4, Grin2a–Grin2b–Slc1a2, and Grin2b–Mecp2–Cdkl5–Tet3.

## GSEA in the environmental comparison

To biologically interpret the transcriptomic alterations, gene lists ranked by their corresponding $\log_2$FC values were used to perform GSEA. The directionality of GO enrichment was evaluated using the $\log_2$FC values as a ranking metric. All GSEA results obtained using clusterProfiler are provided in S7 Data. The dot plots in Fig 8 show the GO-GSEA results focusing on the environmental conditions in the left and right CA1 regions. The analyses identified core enrichment genes, defined as the subset of genes that contributed most significantly to the enrichment score and potentially act as key drivers of the biological changes within each gene set (see S7 Data).

In total, 1,293 gene sets were enriched in the left ISO–ENR analysis. Fig 8A illustrates the GO-GSEA results for the left CA1 region, showing the top 10 gene sets ranked by GeneRatio. Gene sets related to mitochondria and their functions showed a high degree of enrichment in the activated direction, indicating enrichment under ENR conditions. In the suppressed direction, corresponding to enrichment in the ISO condition, gene sets related to synapses (e.g., asymmetric synapses, postsynaptic density) and DNA-binding transcription activator activity were observed. Contrastingly, 496 gene sets were enriched in the right ISO–ENR analysis. Fig 8B shows the GO-GSEA results in the right CA1 region, also presenting the top 10 gene sets ranked by GeneRatio. Gene sets associated with protein folding and RNA splicing were strongly enriched under the ENR conditions. Under the ISO conditions, gene sets related to synapses and transcription activator activity were enriched, consistent with the findings in the left CA1 region. Separate analyses were conducted for the GO-BP, GO-CC, and GO-MF categories. The corresponding dot plots are presented in S5 Fig (GO-BP), S6 Fig (GO-CC), and S7 Fig (GO-MF).

Finally, pathway-based enrichment analyses were performed using MSigDB's C2 Reactome and KEGG databases to complement the GO-based results. These results are shown in S8 Fig (Reactome) and S9 Fig (KEGG). The numbers of significant pathway gene sets are as follows: Reactome, 197 on the left and 28 on the right; and KEGG, 93 on the left and 29 on the right (S7 data). These results strengthen the GO enrichment features described above. In the ENR condition, the left CA1 region showed activation of mitochondrial respiration, and the right CA1 region showed enrichment of RNA splicing. Under the ISO conditions, both regions showed stimulations of synaptic development and function.

These GSEA results complement the ORA findings, suggesting that the left CA1 response under the ENR condition involves enhanced mitochondrial metabolic activity, whereas the right CA1 response under the same condition is characterized by post-transcriptional regulation through RNA splicing. Notably, synaptic processes were commonly enriched in the ISO condition across both hemispheres, further reinforcing the ORA results.

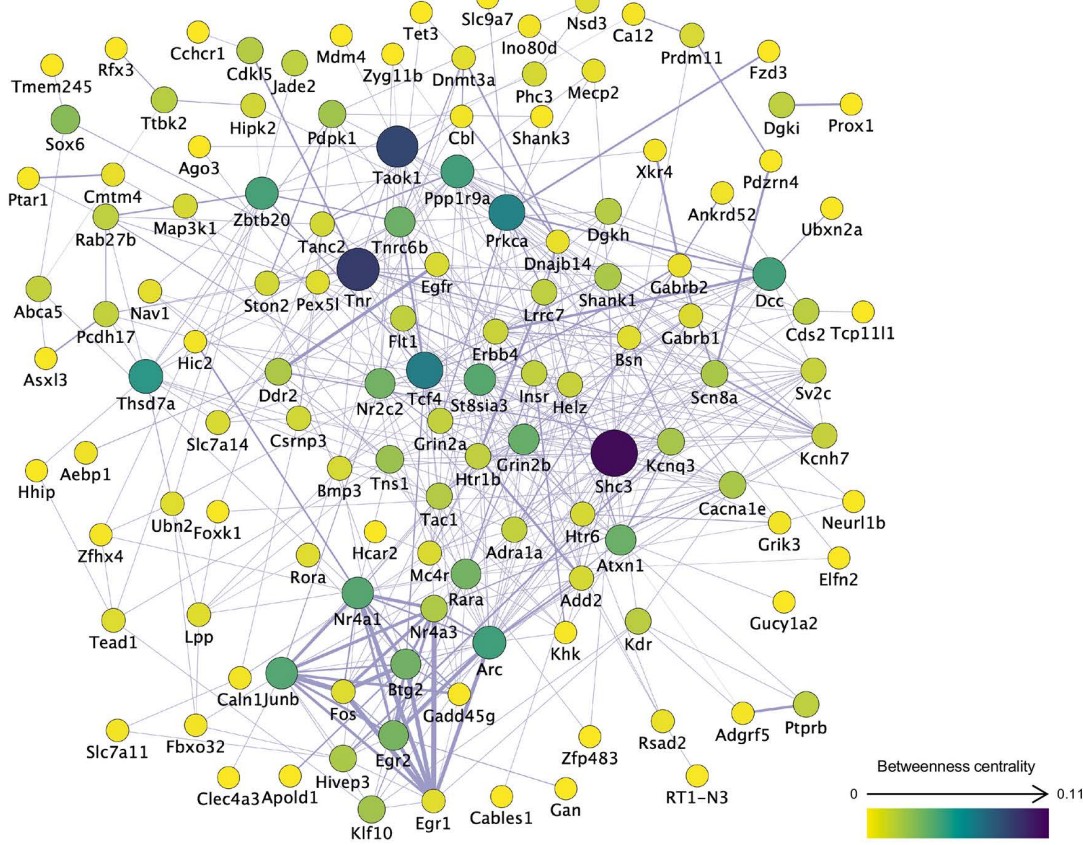
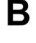
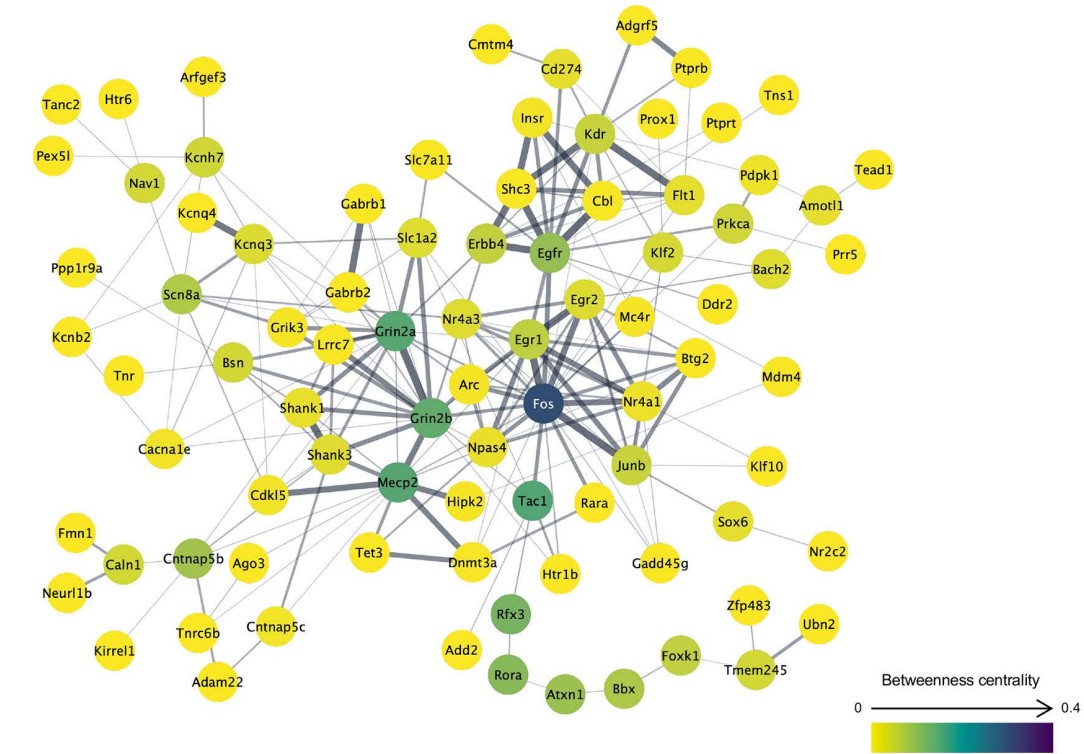

**Fig 5. Gene co-expression and protein–protein interaction networks of the left CA1 region DEGs in the ISO-ENR comparison.** A total of 180 DEGs annotated with gene symbols were included in the analysis. In network visualization, node size and color represent betweenness centrality, with larger and more intensely colored nodes indicating higher centrality. Edge thickness represents the normalized maximum weight score, which reflects the strength of the association between connected genes or proteins. Isolated nodes were manually excluded to ensure visual clarity. (A) Gene co-expression network constructed using the GeneMANIA database. (B) Protein–protein interaction network constructed using the STRING database. ISO, isolated housing conditions; ENR, enriched housing conditions; DEGs, differentially expressed genes.

## Discussion

This study aimed to elucidate how housing conditions influence gene expression in the hippocampal CA1 region, revealing not only distinct transcriptomic landscapes but also common molecular frameworks in the left and right hemispheres. Through comprehensive analyses of DEGs, including their numbers, enrichment profiles, and co-expression/PPI networks, we delineated both unique and shared molecular responses of the bilateral CA1 regions to environmental stimuli.

### Transcriptomic landscapes of the bilateral CA1 regions in response to housing environments

While a direct left–right comparison revealed high baseline similarity (0–1 DEGs), the subsequent ORA and GSEA analyses of environmental responses (ISO vs. ENR) uncovered coherent and biologically distinct functional themes in each hemisphere. These findings suggest that the observed qualitative asymmetry represents a genuine biological response rather than a statistical artifact. This interpretation holds even though the overall $\log_2$FC values of all tested genes were moderately correlated between hemispheres (Pearson's coefficient: $r = 0.587$, S4 Fig), indicating that despite a shared overall trend, the subset of most significantly regulated genes diverged into functionally distinct pathways. In the comparison between the housing conditions, the number of DEGs in the right CA1 region was 94, whereas that in the left CA1 region was 189 (Fig 1). This suggests that the degree of gene expression changes in response to the housing environment differs markedly between the two hemispheres. To further interpret these differences, we performed an ORA on the DEGs from both the left and right hippocampal CA1 regions (ISO vs. ENR conditions), which revealed distinct biological processes and molecular mechanisms of each group. Notably, both regions were enriched in the transcription factors E2F and miR-132-3p, suggesting shared transcriptional and post-transcriptional control mechanisms in response to environmental stimuli in the CA1 region [44,45].

Despite these similarities, significant differences were observed between the two groups. In the left CA1 region, GO terms related to glutamate receptor activity, synapse function, and long-term memory (e.g., glutamate-gated calcium ion channel activity and long-term memory), along with the MAPK signaling pathway, were significantly enriched (Fig 3A). These findings suggest that the activation of gene groups is crucial for neuroplasticity and synaptic transmission, supporting the notion that environmental stimuli predominantly activate mechanisms related to nervous system function and memory formation in the left hemisphere [46]. Conversely, the right CA1 region exhibited enrichment in pathways associated with RNA metabolism and cellular structural reorganization (e.g., regulation of RNA metabolic processes and cell periphery), as illustrated in Fig 3B. Furthermore, the pathways regulated by the neuron-specific transcription factor NPAS4, known for controlling activity-dependent gene expression [47], were enriched, highlighting responses primarily involved in intrinsic cell regulation and homeostasis and suggesting a focus on gene expression regulation in the right hemisphere response.

Next, we employed GSEA to evaluate functional trends at the gene set level in both the left and right hippocampal CA1 regions ISO–ENR groups. The results provided deeper insights into the molecular signatures of the two groups (Fig 8, S8 Fig, and S9 Fig). In the left CA1 region, metabolic activities, including the mitochondrial respiratory chain, oxidative phosphorylation, and cellular respiration, were significantly activated by environmental enrichment. These functional categories primarily focus on energy production and align with the previously observed activation of synaptic transmission, long-term memory, and MAPK pathways (Fig 3A), suggesting that increased neural activity is linked to increased metabolic demand.

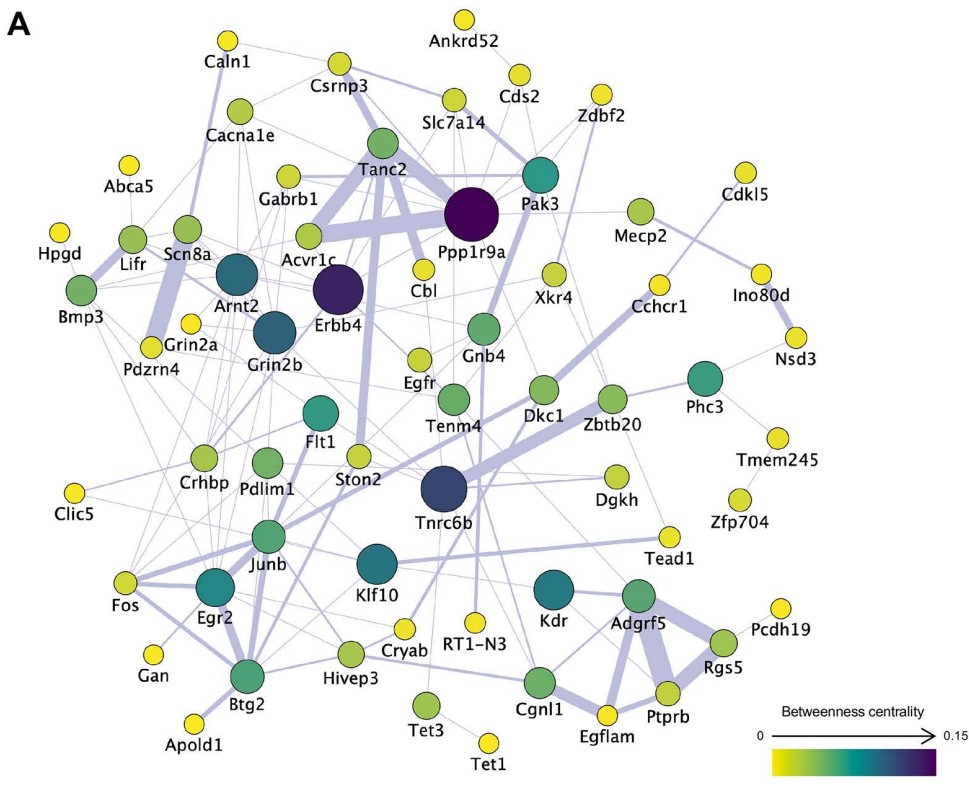

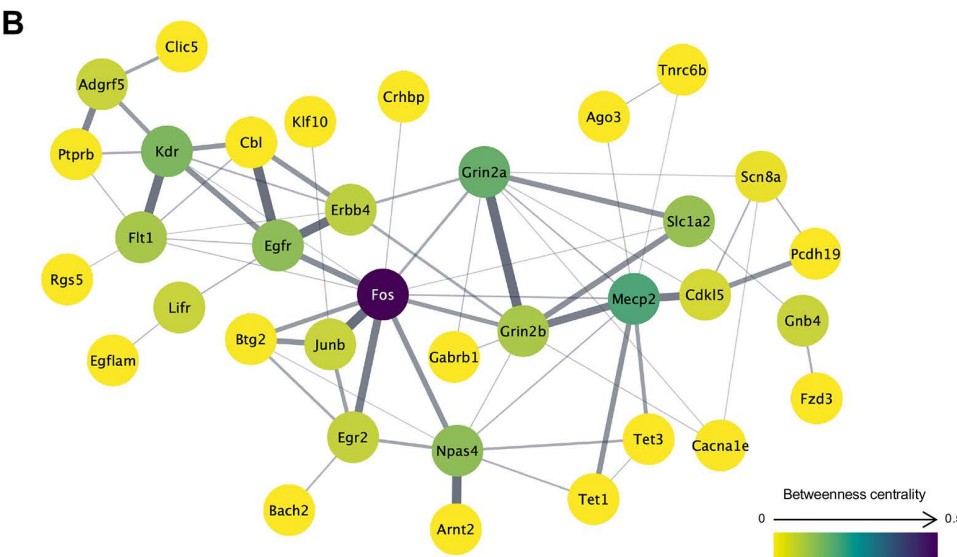

**Fig 6. Gene co-expression and protein–protein interaction networks of the right CA1 region DEGs in the ISO-ENR comparison.** A total of 87 DEGs annotated with gene symbols were included in the analysis. In network visualization, node size and color represent betweenness centrality, with larger and more intensely colored nodes indicating higher centrality. Edge thickness represents the normalized maximum weight score, which reflects the strength of the association between connected genes or proteins. Isolated nodes were manually excluded to ensure visual clarity. (A) Gene co-expression network constructed using the GeneMANIA database. (B) Protein-protein interaction network constructed using the STRING database. ISO, isolated housing conditions; ENR, enriched housing conditions; DEGs, differentially expressed genes.

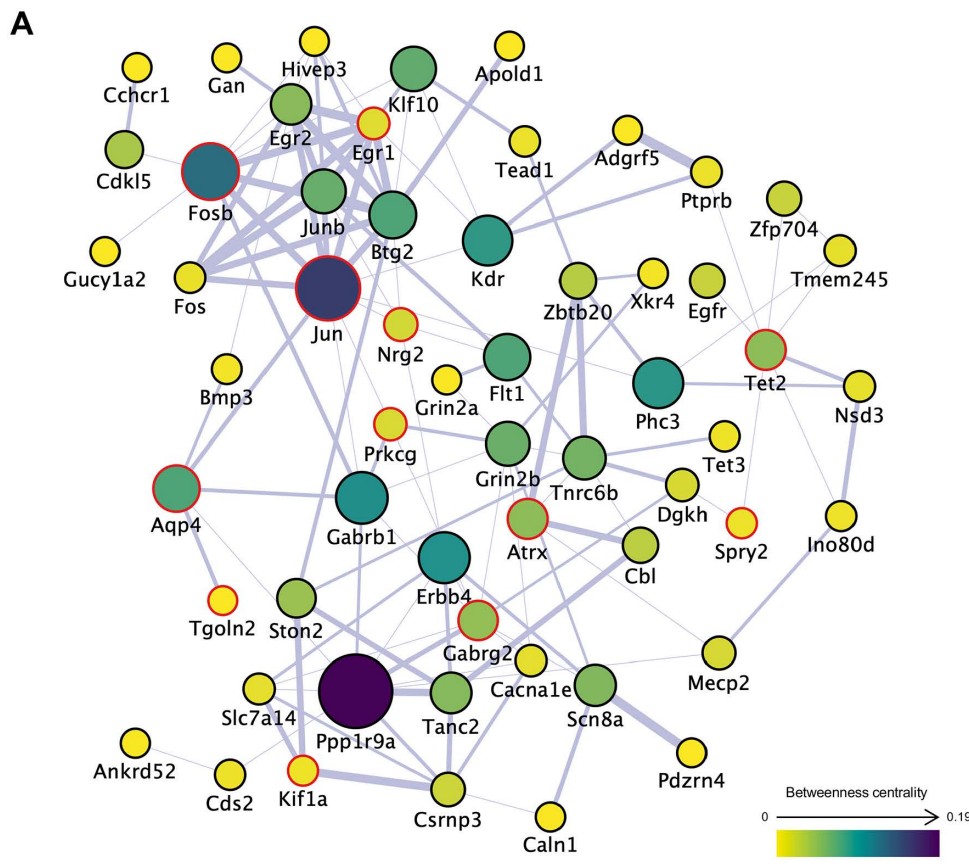

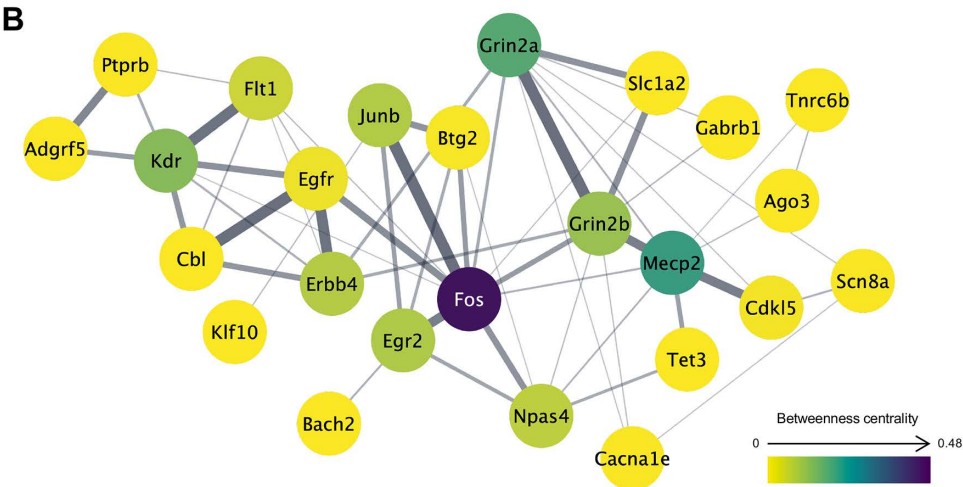

**Fig 7. Gene co-expression and protein–protein interaction networks of bilaterally shared ISO-ENR DEGs.** A total of 64 DEGs annotated with gene symbols were included in the analysis. In network visualization, node size and color represent betweenness centrality, with larger and more intensely colored nodes indicating higher centrality. Edge thickness represents the normalized maximum weight score, which reflects the strength of the association between connected genes or proteins. Isolated nodes were manually excluded to ensure visual clarity. (A) Gene co-expression network constructed using the GeneMANIA database. A query gene-based weighting algorithm was applied to highlight highly correlated genes (marked with red borders) within the network, thereby emphasizing common features among shared DEGs. (B) Protein–protein interaction network constructed using the STRING database. ISO, isolated housing conditions; ENR, enriched housing conditions; DEGs, differentially expressed genes.

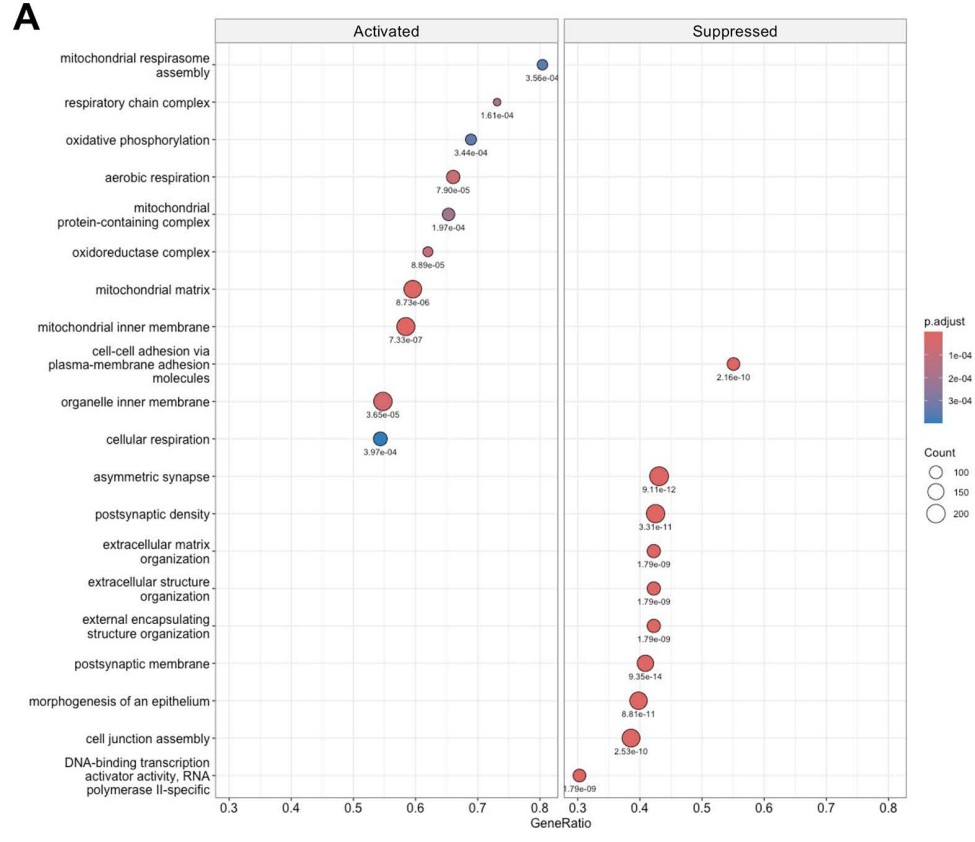

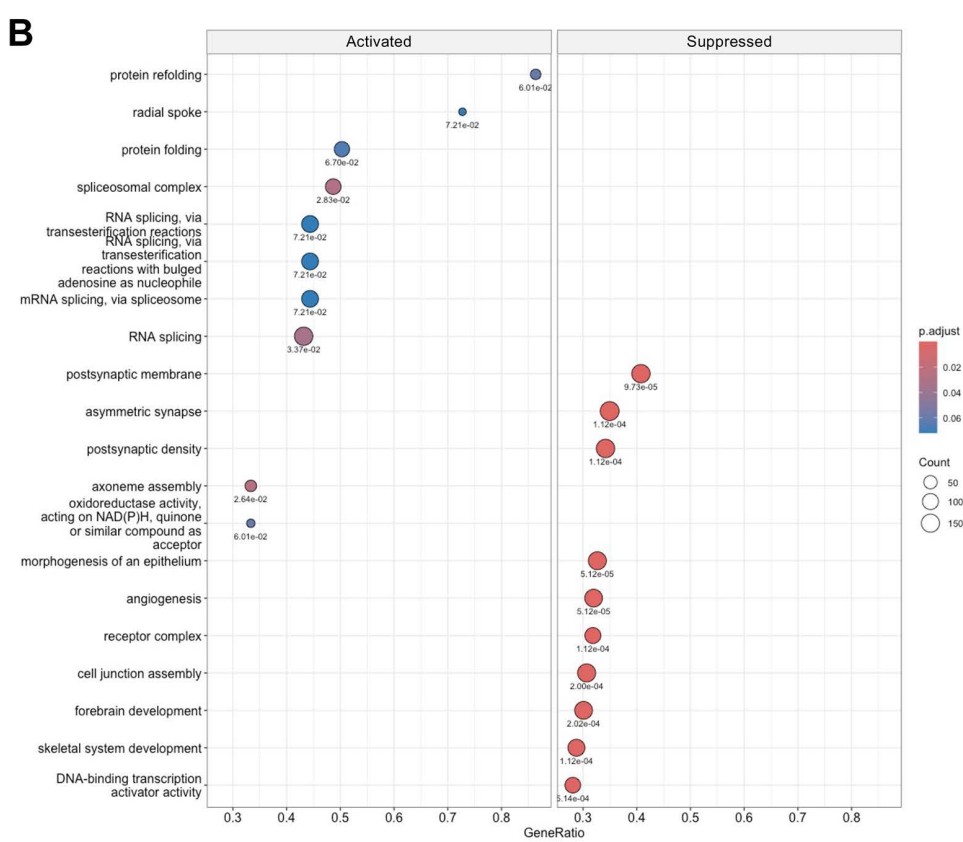

**Fig 8. Biological profiles of environmental conditions based on gene set enrichment analysis (GSEA).** In the ISO–ENR comparison, GSEA using Gene Ontology terms was conducted with $\log_2$(fold change) values from the edgeR analysis as input. Adjusted *p*-values are shown below each dot. The x-axis of the dot plot represents GeneRatio, which is defined as the proportion of genes within a gene set that belong to the core enrichment subset. (A) Dot plot for the left ISO–ENR. (B) Dot plot for the right ISO–ENR. ISO, isolated housing conditions; ENR, enriched housing conditions.

In contrast, post-transcriptional regulations within the nucleus and cytoplasm, such as RNA splicing and protein refolding, were selectively activated in the right CA1 region. This finding aligns with the activation of transcription factors such as NPAS4 and E2F, as shown in Fig 3B, reflecting a central role for intracellular regulatory mechanisms in response to stimuli. Notably, we observed similar enrichment patterns of asymmetric synaptic structures and increased postsynaptic density gene sets under the ISO conditions in both the left and right CA1 regions. Social isolation during adolescence increases gene and protein expression of NR2B in the rodent hippocampus [48,49]. This trend may reflect the significantly higher expression of *Grin2b*, which encodes GluN2B (NR2B), in the bilateral ISO group.

Collectively, these ORA and GSEA results consistently demonstrate that distinct molecular mechanisms underpin the effects of environmental treatments on the left and right hippocampal CA1 regions. The left CA1 region appears to promote structural plasticity, such as the remodeling of synaptic ultrastructure, through coordinated metabolic activity, whereas the right CA1 region preferentially engages endogenous gene regulatory pathways that may suppress experience-driven neural plasticity under enriched environmental conditions. These findings are in line with functional evidence demonstrating that environmental experience enhances interhemispheric asymmetry in hippocampal gamma oscillations [9]. The molecular pathways identified in this study may therefore constitute a biological substrate for such experience-dependent functional lateralization.

### Commonality of environment-induced differential gene expression between the left and right CA1 regions

Co-expression and PPI network analyses of environmentally responsive DEGs in the left and right CA1 regions revealed a shared molecular framework in both areas (Figs 5A, 6A, and 7A). Within the co-expression network, IEGs, such as *Fos*, *Junb*, *Egr1*, and *Btg2,* formed central clusters in both left and right CA1 regions. This suggests that rapid transcriptional responses to environmental stimuli and early memory formation are common initial responses in both hemispheres [50,51].

Beyond IEGs, in the PPI network, molecules involved in neural development and plasticity, specifically Grin2b, Mecp2, Cdkl5, and Tet3, formed functionally connected clusters in both the left and right CA1 regions (Figs 5B, 6B and 7B). This Grin2b–Mecp2–Cdkl5–Tet3 cluster has been implicated in the maintenance of synaptic function, epigenetic regulation, and neurodevelopmental disorders [52–54], and likely constitutes a core basis for environment-dependent gene regulation common to both the left and right CA1 regions. Indeed, our ORA results (Fig 3) showed a significant enrichment of GO items related to synaptic structure in the left CA1 region and transcriptional regulation, including DNA 5-methylcytosine activity in the right CA1 region. The network centered on this cluster may function as a molecular interface that integrates the responses of the two hemispheres through epigenetic control-based neural regulation. The GSEA results using the Reactome Pathway partly support this speculation: the left CA1 region shows enrichment of the gene set "transcriptional regulation by MECP2" in the ISO group (S8 Fig). However, further studies are needed to confirm this hypothesis.

Furthermore, the ORA of the shared DEGs between the left and right CA1 regions in response to environmental exposure (Fig 4) revealed the enrichment of terms related to transcriptional regulation, NMDA receptor activity, synaptic structure, and cellular signaling pathways. Notably, categories such as "sequence-specific DNA binding" and "transcription regulator activity" underscore the activation of shared transcriptional networks involving IEGs such as *Fos, Junb*, and *Egr1*, observed in both co-expression and PPI network analyses. These results indicate the presence of a common transcriptional response module in response to environmental stimuli in both hemispheres. Enrichment of terms such

as "NMDA glutamate receptor activity" and "presynaptic membrane" further supports the shared activation of neural plasticity-related pathways, consistent with the GSEA results showing synaptic-related processes. Moreover, the detection of pathways such as the "calcium signaling pathway" and the involvement of miR-132-3p—both known to be linked to activity-dependent synaptic remodeling and long-term transcriptional control [45,55]—suggest that core molecular mechanisms are symmetrically recruited in experience-dependent circuit reorganization across the bilateral CA1 regions.

In summary, while the left and right CA1 regions employ distinct molecular strategies (e.g., structural vs. transcriptional emphasis), they may share a common foundational response. This shared basis is evident in acute IEG responses and overlapping networks integral to neuroplasticity and epigenetic regulation. Therefore, brain laterality should not be considered a complete dichotomy but rather a graded asymmetry, where diverse response patterns are superimposed on a shared core mechanism. These findings, together with earlier network-based analyses, highlight that despite hemispheric asymmetries, a conserved molecular framework likely orchestrates transcriptional and synaptic responses to environmental input.

## Limitations and perspectives

Careful consideration of sex-specific differences in lateralization is required. Following a previous study, we used male rats to exclude the potential influence of female sex hormones on lateralization [9]. However, hippocampal asymmetry has also been observed in female rats [56,57]. Recent studies have revealed sex-related differences in NMDA receptor function and synaptic plasticity [58,59]. For example, estrogen modulates NMDA receptor signaling and long-term potentiation in females, whereas males may rely more on GluN2B-mediated pathways [60]. These molecular differences may underlie the distinct patterns of hippocampal asymmetry and memory processing between sexes, highlighting the importance of including both sexes in future studies.

Although the present study used the 4-week period following weaning as the window for environmental exposure, the appropriateness of this timeframe remains unclear. Postnatal brain development, including that of the hippocampus, involves a critical period that profoundly influences learning and experience [61,62]. Previous studies on mice have also examined the relationship between the duration of environmental enrichment and subsequent structural and functional changes, as well as alterations in hippocampal gene expression [18]. To gain a more comprehensive understanding, future research should incorporate cross-sectional studies across diverse developmental stages, including old age, and longitudinal studies spanning multiple life phases.

The statistical power of the current study for detecting DEGs may be insufficient because each group included only three animals. Experimental validation using RT-qPCR was performed for some DEGs; however, more detailed and robust statistical analyses would require larger sample sizes. For example, in the left–right comparison, only one DEG, *Pkml1*, was identified in the ISO group. The limited sample size raises concerns about reduced sensitivity; therefore, it is difficult to determine based on this study alone whether the limited detection of significant genes in the left–right comparison is due to insufficient statistical power or to underlying biological factors. Additionally, to compensate for the reduced sensitivity due to the small sample size, a less stringent statistical criterion was used for DEG detection, raising concerns about an increased false-positive rate. Thus, large-scale studies with adequate statistical power are essential for robust and reliable identification of lateralized and environment-induced gene expression profiles.

As suggested by the enrichment analysis results, future investigations on the molecular-level left–right differences should extend beyond the mRNA layer. NMDA receptor subunit genes, including *Grin2b*, contribute to neural functional diversity and precise formation and maintenance of neural circuits through splice variation arising from alternative exon usage [63]. Additionally, *Mecp2*, which was highlighted by network analysis, is involved not only in epigenetic regulation in the brain but also in the regulation of alternative splicing [64]. Furthermore, isoform-specific alterations in Mecp2 protein levels have been implicated in Rett syndrome [65]. To further advance our findings, the identification of novel isoforms through long-read sequencing technologies and the accurate quantification of splice variants using deep sequencing, together with genome-wide epigenetic profiling, is required.

Finally, this study employed bulk RNA-seq analysis of the entire CA1 region. While this approach provides an overall transcriptomic profile, it inherently averages gene expression across diverse cell populations and therefore cannot distinguish between gene expression changes occurring within specific cell types (e.g., pyramidal neurons or astrocytes) and those arising from alterations in cellular composition. Future studies utilizing single-cell RNA sequencing (scRNA-seq) or spatial transcriptomics will be essential to elucidate cell-type-specific and spatially resolved responses to environmental stimuli in the bilateral hippocampus.

## Supporting information

**S1 Fig. Brief workflow chart of this study.**
(PDF)

**S2 Fig. Multivariate analyses using the TPM-normalized gene count data.**
(PDF)

**S3 Fig. Volcano plot visualizations of the left–right comparison.**
(PDF)

**S4 Fig. Correlation plots focusing on the $log_2$FC values calculated using edgeR.**
(PDF)

**S5 Fig. GSEA for the environmental comparisons in the left and right CA1 regions using the GO-BP database.**
(PDF)

**S6 Fig. GSEA for the environmental comparisons in the left and right CA1 regions using the GO-CC database.**
(PDF)

**S7 Fig. GSEA for the environmental comparisons in the left and right CA1 regions using the GO-MF database.**
(PDF)

**S8 Fig. GSEA for the environmental comparisons in the left and right CA1 regions using the MSigDB's C2 Reactome Pathway database.**
(PDF)

**S9 Fig. GSEA for the environmental comparisons in the left and right CA1 regions using the MSigDB's C2 KEGG Pathway database.**
(PDF)

**S1 Table. Primer information used in RT-qPCR.**
(XLSX)

**S1 Data.  Raw and TPM-normalized gene count data.**
(XLSX)

**S2 Data. The edgeR calculation results of the left–right comparison.**
(XLSX)

**S3 Data.  The edgeR calculation results of the ISO–ENR comparison.**
(XLSX)

**S4 Data. The DEG lists of the ISO-ENR comparison.**
(XLSX)

**S5 Data. The g:Profiler calculation results of over-representation analysis.**
(XLSX)

**S6 Data. The Cytoscape calculation results of network analysis.**
(XLSX)

**S7 Data. The clusterProfiler calculation results of gene set enrichment analysis.**
(XLSX)

## Acknowledgments

The authors thank Dr. Takatoshi Ueki and Dr. Hajime Hirase for their kind advice. The authors thank Ms. Yoko Iwauchi for technical assistance. We would like to thank Editage (www.editage.jp) for English language editing.

## Author contributions

**Conceptualization:** Yoshiaki Shinohara, Atsushi Tajima.

**Data curation:** Kentaro Kojima, Takayuki Kannon, Yoshiaki Shinohara.

**Formal analysis:** Azusa Kubota, Takehiro Sato.

**Funding acquisition:** Azusa Kubota, Yoshiaki Shinohara, Atsushi Tajima.

**Investigation:** Azusa Kubota, Kentaro Kojima, Shinnosuke Koketsu, Kazuyoshi Hosomichi.

**Resources:** Yoshiaki Shinohara, Atsushi Tajima.

**Writing – original draft:** Azusa Kubota, Atsushi Tajima.

**Writing – review & editing:** Azusa Kubota, Kentaro Kojima, Shinnosuke Koketsu, Takayuki Kannon, Takehiro Sato, Kazuyoshi Hosomichi, Yoshiaki Shinohara, Atsushi Tajima.

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
