## [Decision Letter · Decision Letter 0]

17 Jun 2025

Dear Dr. TAJIMA,

Thank you for submitting your manuscript to PLOS ONE. After careful consideration, we feel that it has merit but does not fully meet PLOS ONE’s publication criteria as it currently stands. Therefore, we invite you to submit a revised version of the manuscript that addresses the points raised during the review process.

We look forward to receiving your revised manuscript.

Kind regards,

Zhiheng Lin, (Ph.D., M.D.

Academic Editor

PLOS ONE

Journal Requirements:

2. To comply with PLOS ONE submissions requirements, in your Methods section, please provide additional information regarding the experiments involving animals and ensure you have included details on (1) methods of sacrifice, and (2) efforts to alleviate suffering.

“This work was partially supported by JSPS KAKEN Grant Numbers JP17H02221 and JP20H03295 (to YS), and JP23K18386 and JP23H03145 (to AT). This work was also supported by JST SPRING, Grant Number JPMJSP2135 (to AK). This work included results obtained using shared equipment under the MEXT Project for promoting public utilization of advanced research infrastructure (Program for supporting construction of core facilities) Grant Number JPMXS0440300024.

4. Thank you for stating the following in your manuscript:

“This work was partially supported by JSPS KAKEN Grant Numbers JP17H02221 and JP20H03295 (to YS), and JP23K18386 and JP23H03145 (to AT). This work was also supported by JST SPRING, Grant Number JPMJSP2135 (to AK). This work included results obtained using shared equipment under the MEXT Project for promoting public utilization of advanced research infrastructure (Program for supporting construction of core facilities) Grant Number JPMXS0440300024.”

“This work was partially supported by JSPS KAKEN Grant Numbers JP17H02221 and JP20H03295 (to YS), and JP23K18386 and JP23H03145 (to AT). This work was also supported by JST SPRING, Grant Number JPMJSP2135 (to AK). This work included results obtained using shared equipment under the MEXT Project for promoting public utilization of advanced research infrastructure (Program for supporting construction of core facilities) Grant Number JPMXS0440300024.”

Reviewers' comments:

Reviewer's Responses to Questions

**Comments to the Author**

1. Is the manuscript technically sound, and do the data support the conclusions?

Reviewer #1: Yes

Reviewer #2: Partly

2. Has the statistical analysis been performed appropriately and rigorously?

Reviewer #1: Yes

Reviewer #2: Yes

3. Have the authors made all data underlying the findings in their manuscript fully available?

Reviewer #1: Yes

Reviewer #2: No

4. Is the manuscript presented in an intelligible fashion and written in standard English?

Reviewer #1: No

Reviewer #2: Yes

Reviewer #1: The current manuscript mainly involves the environmental regulation of hippocampal lateralization, but there are several problems that limit its scientific nature, and major modifications are recommended.

1. In the abstract, it is recommended to introduce the effects of environment on neuroanatomy and transcriptome in detail and compare with the existing research results. Especially on the impact of the environment on the left and right CA1 regions, it is recommended to increase the relevant literature support.

2.For data analysis, especially the description of statistical methods is relatively simple, and the lack of specific explanation for the selection of key parameters, it is recommended to describe the statistical methods of data analysis in detail.

3.There is a lack of in-depth explanation in the results, especially for the 'no DEG detected ' part; it is recommended to increase the discussion to explain the possible reasons.

4. It is suggested to discuss possible mechanisms, such as differences in neuroanatomy and function between left and right CA1, or how environmental stimuli affect gene expression through stress response.

5. It is recommended that the function and underlying mechanisms of each key gene be discussed in detail, including those genes that may not be expressed significantly in the DEG analysis, as they may play a role in environmental stress.

6.The limitations of the article were discussed in more detail, especially the influence of gender differences on the heterogeneity of left and right CA1.

7.Slightly simplify some complex sentences to ensure that they are concise and clear.

8.Improve the logic of the article and avoid repeated statements.

9. The references cited in this article are not sufficient, and there is a lack of in-depth comparative discussion. Background and methodology also require further literature support. Some related research should be cited:

1. Zn‐DHM Nanozymes Enhance Muscle Regeneration Through ROS Scavenging and Macrophage Polarization in Volumetric Muscle Loss Revealed by Single‐Cell Profiling

2. An injectable multi-functional composite bioactive hydrogel for bone regeneration via immunoregulatory and osteogenesis effects

3. Single-cell RNA sequencing and immune microenvironment analysis reveal PLOD2-driven malignant transformation in cervical cancer

4. Single-cell analysis unveils cell subtypes of acral melanoma cells at the early and late differentiation stages

5. The cellular signaling crosstalk between memory B cells and tumor cells in nasopharyngeal carcinoma cannot be overlooked: Their involvement in tumor progression and treatment strategy is significant

6. Single-Cell RNA-Sequencing Integration Analysis Revealed Immune Cell Heterogeneity in Five Human Autoimmune Diseases

7. Prognostic Value of Tumor-microenvironment-associated Genes in Ovarian Cancer

Reviewer #2: This study, based on RNA-seq technology, explored the effects of different living environments on the transcriptome of the left and right CA1 regions of the rat hippocampus, aiming to understand whether the environment regulates the lateralization of brain function. The topic has certain theoretical value, the research design is basically reasonable, the technical route is standardized, and the preliminary conclusion is enlightening. However, the current manuscript has obvious deficiencies in data screening, database usage, result interpretation, and logical argumentation. It is recommended to make substantial revisions (major revision) to this manuscript. On the basis of improving data support and conclusion logic, further enhance the quality and publication value of the manuscript.

1. The authors set a relatively strict DEG screening standard (log2FC > ±0.58, FDR < 0.05), resulting in a small number of differentially expressed genes being screened out: only 13 on the right CA1 and only 105 on the left CA1. This number is insufficient to support an in-depth interpretation of the transcriptional response of complex brain regions under environmental intervention. It is recommended to supplement the use of a more lenient threshold (such as FDR < 0.1 or p < 0.05) for trend differential expression analysis and provide MA plots, rank plots, or a complete list of gene expression logFC to show the overall expression trend or clearly explain the statistical power and limitations of the current results, avoiding over-interpretation of the results.

2. Some of the tools used in the current study (such as Salmon v0.8.1, GeneMANIA, and org.Rn.eg.db v3.20) are significantly outdated, which may affect the accuracy and reproducibility of the analysis. It is recommended to upgrade Salmon to v1.10 or above, supplement bias correction and other functions, and annotate the used Ensembl version, or use the latest GENCODE or Ensembl v110. In the GSEA part, it is recommended to introduce the MSigDB database (through the msigdbr package) to enhance the breadth and interpretability of the GO enrichment results. For co-expression analysis, it is recommended to combine the STRING database or use more robust analysis tools such as WGCNA.

3. The current discussion provides relatively brief mechanistic explanations for several key findings and fails to fully integrate existing research for in-depth exploration. It is recommended to deeply discuss the functions of key genes (such as Fos, Cdkl5, Grin2b, etc.) in neural activity and analyze the biological significance of pathways such as synaptic transmission and cell connections based on GSEA enrichment pathways. In the conclusion section, the language should be more cautious, clearly stating that the study is a preliminary exploration.

4. With only three biological replicates per group, it may not be sufficient to support the detection of DEGs in some inter-group comparisons (such as left and right CA1). It is recommended to supplement statistical power assessment or point out this limitation in the discussion.

5. This study has potential value, but the current data support and argumentation process are not yet complete. It is recommended to deepen the expression analysis, clarify the analysis methods and versions and the scope and limitations of the results, and strengthen the biological interpretation and mechanism exploration.

6. It is recommended to discuss in the discussion the possible influence of gender factors on the lateralized expression of CA1 and its necessity in future research. In addition, there are some minor suggestions, such as adding an experimental flowchart, making the chart annotations clearer, unifying the abbreviations of terms, and using language polishing services to improve the quality of English.

The references cited in this article are not sufficient, and there is a lack of in-depth comparative discussion. Background and methodology also require further literature support. Some related research should be cited:

10.1111/cpr.13376

10.18240/ijo.2019.08.04

10.34133/research.0183

10.34133/research.0134

**Do you want your identity to be public for this peer review?** For information about this choice, including consent withdrawal, please see our Privacy Policy

Reviewer #1: No

Reviewer #2: No

---

## [Author Response · Author response to Decision Letter 1]

26 Aug 2025

Response to Reviewers (PONE-D-25-08336)

To Reviewer #1

1. In the abstract, it is recommended to introduce the effects of environment on neuroanatomy and transcriptome in detail and compare with the existing research results. Especially on the impact of the environment on the left and right CA1 regions, it is recommended to increase the relevant literature support.

Response:

Thank you for this helpful suggestion. After careful consideration, we decided that revising the Introduction section of the manuscript would be more effective than modifying the Abstract. Specifically, we have rewritten the paragraph introducing the effects of the environment on neuroanatomy and the transcriptome (lines 72–92) to draw more support from the relevant literature. Additionally, we have increased the number of citations to strengthen the supporting evidence from previous studies, increasing the total from 36 to 65.

----------

2. For data analysis, especially the description of statistical methods is relatively simple, and the lack of specific explanation for the selection of key parameters, it is recommended to describe the statistical methods of data analysis in detail.

Response:

Thank you for your comments. Given that the specific part lacking details was not specified, we have added further explanations to enhance the description of the analytical methods in the Materials and Methods section of the revised manuscript (lines 156–161, 180–184, 193–198, and 206–215).

----------

3. There is a lack of in-depth explanation in the results, especially for the 'no DEG detected ' part; it is recommended to increase the discussion to explain the possible reasons.

Response:

Thank you for your valuable comments. In this revised manuscript, we adopted less stringent thresholds in response to Comment 1 from Reviewer #2. We identified one differentially expressed gene (DEG) in the left–right comparison within the ISO group. Although no DEGs were detected in the ENR group, even with the relaxed thresholds, we have added a discussion of the possible reasons why the present study did not reveal notable hemispheric differences in gene expression (lines 554–565).

----------

4. It is suggested to discuss possible mechanisms, such as differences in neuroanatomy and function between left and right CA1, or how environmental stimuli affect gene expression through stress response.

Response:

Thank you for the important suggestion. We agree with your suggestion. Accordingly, we have substantially revised the Discussion section to introduce the possible mechanisms through which living environments may contribute to hippocampal asymmetry. For example, in lines 482–490, we described the proposed role of transcriptomic differences in generating hippocampal asymmetry: “Collectively, these ORA and GSEA results consistently demonstrate that distinct molecular mechanisms underpin the effects of environmental treatments on the left and right hippocampal CA1 regions. While the left CA1 region facilitates coordinated metabolic activity and structural remodeling, the right CA1 region selectively activates endogenous gene regulatory mechanisms, potentially suppressing experience-driven neural plasticity in response to enriched environments. These findings highlight qualitative differences in the molecular strategies employed by each hemisphere in response to environmental stimuli, offering a detailed perspective on functional asymmetry and localized response specificity in the hippocampus.”

----------

5. It is recommended that the function and underlying mechanisms of each key gene be discussed in detail, including those genes that may not be expressed significantly in the DEG analysis, as they may play a role in environmental stress.

Response:

Thank you for your recommendations. In the revised manuscript, we have expanded the discussion of specific DEGs to strengthen the interpretation of our findings. As mentioned previously (in response to comment 3), the use of a new FDR threshold revealed numerous candidate DEGs that warranted further discussion in the context of hippocampal asymmetry and environmental influences on gene expression profiles. We believe that the current revision, particularly the updated discussion on the role of the environment (e.g., lines 491–534), addresses the concerns raised in the previous version of our manuscript.

----------

6. The limitations of the article were discussed in more detail, especially the influence of gender differences on the heterogeneity of left and right CA1.

Response:

Thank you for the suggestion. We have explained the limitations of our study in more detail, including the potential influence of sex differences on hippocampal laterality. Particularly, we introduced a new subsection titled “Limitations and perspectives” in the Discussion section to clarify how our findings may contribute to future research and which aspects remain inconclusive based on the current study.

----------

7. Slightly simplify some complex sentences to ensure that they are concise and clear.

8. Improve the logic of the article and avoid repeated statements.

Response:

Thank you for your suggestions regarding the structure of the manuscript. Comments 7 and 8 reflected a similar perspective. To improve the logical flow and eliminate redundant statements, we have revised the manuscript with the assistance of a language-polishing service. This also helped to enhance the overall quality of the writing. We hope that these revisions have addressed your concerns.

----------

9. The references cited in this article are not sufficient, and there is a lack of in-depth comparative discussion. Background and methodology also require further literature support. Some related research should be cited:

1. Zn‐DHM Nanozymes Enhance Muscle Regeneration Through ROS Scavenging and Macrophage Polarization in Volumetric Muscle Loss Revealed by Single‐Cell Profiling

2. An injectable multi-functional composite bioactive hydrogel for bone regeneration via immunoregulatory and osteogenesis effects

3. Single-cell RNA sequencing and immune microenvironment analysis reveal PLOD2-driven malignant transformation in cervical cancer

4. Single-cell analysis unveils cell subtypes of acral melanoma cells at the early and late differentiation stages

5. The cellular signaling crosstalk between memory B cells and tumor cells in nasopharyngeal carcinoma cannot be overlooked: Their involvement in tumor progression and treatment strategy is significant

6. Single-Cell RNA-Sequencing Integration Analysis Revealed Immune Cell Heterogeneity in Five Human Autoimmune Diseases

7. Prognostic Value of Tumor-microenvironment-associated Genes in Ovarian Cancer

The suggested references are not directly relevant to this manuscript. Accordingly, we believe that adding the citations to the manuscript would be inappropriate. Hence, we have decided not to include the suggested references in the manuscript. Thank you for your comments.

----------

To Reviewer #2

1. The authors set a relatively strict DEG screening standard (log2FC > ±0.58, FDR < 0.05), resulting in a small number of differentially expressed genes being screened out: only 13 on the right CA1 and only 105 on the left CA1. This number is insufficient to support an in-depth interpretation of the transcriptional response of complex brain regions under environmental intervention. It is recommended to supplement the use of a more lenient threshold (such as FDR < 0.1 or p < 0.05) for trend differential expression analysis and provide MA plots, rank plots, or a complete list of gene expression logFC to show the overall expression trend or clearly explain the statistical power and limitations of the current results, avoiding over-interpretation of the results.

Response:

We appreciate your valuable comment and agree fully with your suggestions. Accordingly, we adopted a less stringent FDR threshold (relaxed from 0.05 to 0.1). This modification increased the number of environment-induced DEGs, thereby enhancing the informativeness of the enrichment analyses, especially in the over-representation analyses. As a result of this improvement in the DEG analyses, we strengthened the Discussion section of the revised manuscript. Additionally, the complete expression dataset is provided in the Supplementary Information. We have also addressed the issues of sample size and statistical power in the Limitations and Perspectives subsection of the Discussion (lines 554–565).

----------

2. Some of the tools used in the current study (such as Salmon v0.8.1, GeneMANIA, and org.Rn.eg.db v3.20) are significantly outdated, which may affect the accuracy and reproducibility of the analysis. It is recommended to upgrade Salmon to v1.10 or above, supplement bias correction and other functions, and annotate the used Ensembl version, or use the latest GENCODE or Ensembl v110. In the GSEA part, it is recommended to introduce the MSigDB database (through the msigdbr package) to enhance the breadth and interpretability of the GO enrichment results. For co-expression analysis, it is recommended to combine the STRING database or use more robust analysis tools such as WGCNA.

Response:

Thank you for your suggestion regarding the version of the tools used. First, we reanalyzed the sequencing data using the latest version of Salmon (v1.10.3), along with the Ensembl release 113 reference dataset for mRatBN7.2 cDNA assembly and its corresponding genomic annotation. We also confirmed that the validateMapping option, which improves the accuracy of transcript quantification, was enabled during salmon analysis. Second, we incorporated MSigDB and its R package, msigdbr, to further characterize the GSEA results. Specifically, we used the C2 category gene sets from the KEGG and Reactome Pathway collections in this revision. Third, we performed protein–protein interaction analyses using the STRING database to highlight the relationships among environment-induced DEGs at the post-transcriptional level. Based on your suggestion, we are confident that the scientific value of this study has been further enhanced.

----------

3. The current discussion provides relatively brief mechanistic explanations for several key findings and fails to fully integrate existing research for in-depth exploration. It is recommended to deeply discuss the functions of key genes (such as Fos, Cdkl5, Grin2b, etc.) in neural activity and analyze the biological significance of pathways such as synaptic transmission and cell connections based on GSEA enrichment pathways. In the conclusion section, the language should be more cautious, clearly stating that the study is a preliminary exploration.

Response:

We appreciate your recommendations regarding the Discussion section. As the present study was improved by adopting new DEG thresholds, we have made extensive revisions to this section. Additional mentions and discussions of specific DEGs, including Fos, Cdkl5, and Grin2b, were incorporated based on the combined results of enrichment and network analyses (lines 491–526). Furthermore, by adding the Limitations and Perspectives subsection to the Discussion, we revised the manuscript to adopt a more cautious tone and clearly acknowledged that the current study represents a preliminary exploration.

We appreciate your helpful suggestions, which have significantly contributed to improving the clarity and scientific depth of our manuscript.

----------

4. With only three biological replicates per group, it may not be sufficient to support the detection of DEGs in some inter-group comparisons (such as left and right CA1). It is recommended to supplement statistical power assessment or point out this limitation in the discussion.

Response:

Thank you for this valuable suggestion. As noted in our response to Comment 3 from Reviewer#1, by applying the new FDR threshold, we identified one DEG in the left–right comparison within the ISO group. Although no DEGs were found in the ENR group, even with less stringent thresholds, we have discussed the possible reasons why our study did not observe notable DEGs between the hemispheres (lines 491–553). As previously addressed in our response to Comment 1 regarding statistical power, we discuss the sample size and its implications for statistical power in the Limitations and Perspectives subsection (lines 554–565).

----------

5. This study has potential value, but the current data support and argumentation process are not yet complete. It is recommended to deepen the expression analysis, clarify the analysis methods and versions and the scope and limitations of the results, and strengthen the biological interpretation and mechanism exploration.

Response:

We appreciate your comment regarding the potential value and current limitations of the interpretation of our study. In response, we have made several key revisions to deepen our biological insight and enhance the rigor of our transcriptomic analyses. First, we conducted overrepresentation analyses to further characterize the unilateral DEGs identified in the environmental comparisons. Second, following your recommendation, we performed protein–protein interaction network analyses using the STRING database. Third, we performed an additional GSEA using MSigDB gene sets to supplement the interpretation of gene expression signatures associated with housing conditions. We believe that these modifications substantially strengthened the transcriptomic analysis and biological interpretation in the revised manuscript. The Materials and Methods section has also been updated to include more detailed information on the versions of software and tools used, as well as specific analytical parameters. Furthermore, we have added a new subsection titled “Limitations and perspectives” to the Discussion section to clarify both the implications of our findings for future research and the conclusions that cannot be drawn from the present data.

----------

6. It is recommended to discuss in the discussion the possible influence of gender factors on the lateralized expression of CA1 and its necessity in future research. In addition, there are some minor suggestions, such as adding an experimental flowchart, making the chart annotations clearer, unifying the abbreviations of terms, and using language polishing services to improve the quality of English.

Response:

Thank you for the suggestion. In response, we have added a discussion on the possible influence of sex-related factors on hippocampal lateralization and emphasized its importance in future research (lines 536–544). We have prepared the brief workflow chart of this study in S1 Fig. Additionally, with the assistance of a professional language editing service, we have improved the overall clarity and readability of the manuscript, revised the figure legends, and ensured consistency in the use of abbreviations throughout the text.

The references cited in this article are not sufficient, and there is a lack of in-depth comparative discussion. Background and methodology also require further literature support. Some related research should be cited:

10.1111/cpr.13376

10.18240/ijo.2019.08.04

10.34133/research.0183

10.34133/research.0134

As noted in our response to Comment 9 from Reviewer #1, the suggested citations fall outside the scope of our study. Therefore, we respectfully chose not to include them and appreciate your understanding.

----------

---

## [Decision Letter · Decision Letter 1]

24 Oct 2025

PLOS ONE

Dear Dr. TAJIMA,

Thank you for submitting your manuscript to PLOS ONE. After careful consideration with the assistance of three additional Reviewers, we feel that it has merit but does not fully meet PLOS ONE’s publication criteria as it currently stands. Therefore, we invite you to submit a revised version of the manuscript that addresses the points raised during the review process.

We look forward to receiving your revised manuscript.

Kind regards,

Stephen D. Ginsberg, Ph.D.

Section Editor

PLOS ONE

Journal Requirements:

Reviewers' comments:

Reviewer's Responses to Questions:

**Comments to the Author**

Reviewer #3: All comments have been addressed

Reviewer #4: (No Response)

Reviewer #5: (No Response)

2. Is the manuscript technically sound, and do the data support the conclusions?

Reviewer #3: Yes

Reviewer #4: Partly

Reviewer #5: No

3. Has the statistical analysis been performed appropriately and rigorously?

Reviewer #3: Yes

Reviewer #4: No

Reviewer #5: No

4. Have the authors made all data underlying the findings in their manuscript fully available?

Reviewer #3: Yes

Reviewer #4: Yes

Reviewer #5: Yes

5. Is the manuscript presented in an intelligible fashion and written in standard English?

Reviewer #3: Yes

Reviewer #4: Yes

Reviewer #5: Yes

Reviewer #3: The manuscript "Housing environment bilaterally alters transcriptomic profile in the rat hippocampal CA1 region" is an interesting one and has been conducted scientifically. The authors have addressed the suggestions of the reviewers.

Reviewer #4: The study analyzes data obtained from only three animals per group, which limits the strength of the conclusions. It does not consider the cellular composition of the CA1 region or how environmental factors might impact this composition and, in turn, gene expression. Greater integration with existing literature is needed to contextualize the findings, as was requested by previous reviewers. This has not been sufficiently addressed.

e.g from reviewer: 4. It is suggested to discuss possible mechanisms, such as differences in neuroanatomy and function between left and right CA1

It is commendable that the Limitations section acknowledges the flaws in the study, demonstrating an awareness of the study’s constraints.

While gene validation was performed using qPCR, the rationale as to the selection of two genes compared to all identified is not addressed. Were others analysed but not validated? There is no figure 2C. The number of animals included in this validation is also not specified. It is also unclear whether the same samples were used or if different ones were analyzed. These issues should be explicitly addressed in the Limitations section.

The rationale for each statistical approach should be clearly articulated. It is important to explain why each test was chosen and why it is appropriate for such small sample sizes. Without this justification, the robustness of the statistical conclusions remains uncertain.

Reviewer #5: In this manuscript, the authors investigated the effects of an enriched environment on gene expression in the hippocampal CA1 region of rats, including an analysis of hemispheric differences. Their direct comparison between the left and right hemispheres revealed very few differentially expressed genes (DEGs). However, when analyzing the effects of the enriched environment, the number of DEGs identified based on the authors' threshold differed between the two hemispheres. The authors then performed gene set enrichment and network analyses on these DEGs.

The conclusions drawn from the results are problematic, and the discussion is insufficient. Substantial revisions are required.

Major Points:

1. As other reviewers may have noted, the interpretation of the results is insufficient. The Results section, in particular, merely describes the findings without any interpretation, making it unclear what conclusions can be drawn from each experiment. The authors should interpret their findings within the Results section or significantly expand the Discussion to connect their data to biological meaning.

2. The authors conclude that there are significant hemispheric differences based on their results. However, it is generally expected that some differences, including experimental noise, will emerge in any omics analysis. This reviewer thinks that a more appropriate conclusion from these data would be that there are few, if any, meaningful differences between the hemispheres. The authors must define the criteria under which they would conclude a lack of hemispheric difference and provide stronger evidence that the observed differences are biologically significant, not just experimental noise or statistical artifacts.

3. It is inappropriate to assess hemispheric differences based solely on whether a gene passes a specific DEG threshold in one hemisphere but not the other. A gene identified as a DEG in one hemisphere might show a similar trend (i.e., similar direction and magnitude of change) in the opposite hemisphere, even if it does not meet the arbitrary p-value or fold-change cutoff. To substantiate the claim that the changes are truly hemisphere-specific, the authors should, for 150 genes, present bar graphs displaying the expression levels and corresponding p-values for both hemispheres side-by-side. This would provide a more transparent and convincing comparison.

4. This paper used an “Isolation” condition as a control compared to an “Enriched environment.” Since isolation eliminates social communication, which is critical for normal mouse development, it is questionable whether this condition serves as a proper neutral baseline. A group housing condition should have been included as a control group. This would allow for a clearer determination of how transcription becomes altered by enriched environment.

Minor Points:

1. Although the authors mention the duration of housing in either the isolated or enriched environment in the Discussion, this information is essential for reproducibility and should be clearly stated in the Materials and Methods section.

2. The volcano plots should be annotated with gene names instead of, or in addition to, gene IDs to improve readability and facilitate immediate interpretation.

**Do you want your identity to be public for this peer review?**  For information about this choice, including consent withdrawal, please see our Privacy Policy

Reviewer #3: **Yes: ** Chandra Sekhar Mukhopadhyay

Reviewer #4: No

Reviewer #5: No

---

## [Author Response · Author response to Decision Letter 2]

7 Nov 2025

Response to Reviewers (PONE-D-25-08336R1)

To Reviewer #3

The manuscript "Housing environment bilaterally alters transcriptomic profile in the rat hippocampal CA1 region" is an interesting one and has been conducted scientifically. The authors have addressed the suggestions of the reviewers.

Response:

We sincerely thank Reviewer #3 for the positive and encouraging feedback on our work.

------

To Reviewer #4

The study analyzes data obtained from only three animals per group, which limits the strength of the conclusions. It does not consider the cellular composition of the CA1 region or how environmental factors might impact this composition and, in turn, gene expression. Greater integration with existing literature is needed to contextualize the findings, as was requested by previous reviewers. This has not been sufficiently addressed.

e.g from reviewer: 4. It is suggested to discuss possible mechanisms, such as differences in neuroanatomy and function between left and right CA1

Response:

We sincerely thank the reviewer for the thoughtful and constructive feedback. We have carefully revised the manuscript to address all the concerns raised, particularly by improving the integration with existing literature and providing a clearer discussion of the study’s limitations.

1. Sample size:

We fully agree that the small sample size represents a critical limitation of our study. In the first-round revised manuscript, we have already acknowledged this issue in the “Limitations and perspectives” section, emphasizing the limited statistical power and the potential risk of both false negatives and false positives (Lines 585–596 in the latest version). We also believe that our previous statement—highlighting that “large-scale studies with adequate statistical power are essential for robust and reliable identification of lateralized and environment-induced gene expression profiles” (Lines 594–596) —adequately addresses this concern.

2. Cellular composition:

We appreciate this important point. We have added a new paragraph to the “Limitations and perspectives” section noting that our bulk RNA-seq cannot distinguish cell-type-specific transcriptional changes from possible shifts in cellular composition. We also indicate that future studies employing single-cell RNA sequencing (scRNA-seq) will be essential to clarify these aspects (Lines 603–608).

3. Integration with existing literature (neuroanatomy and function):

We have expanded the Discussion to provide stronger integration with relevant neuroanatomical and functional studies.

- Neuroanatomy: In the revised Discussion (Lines 512–521), we now link our molecular findings on metabolic activity to neuroanatomical features, suggesting that the left CA1 may utilize higher metabolic activity to fuel structural plasticity, such as the remodeling of synaptic ultrastructure.

- Function: In the same paragraph, we further connect our findings with prior reports on hippocampal functional asymmetry by proposing that the observed molecular asymmetry may represent a potential “molecular substrate” for the lateralized gamma oscillations previously described (Line 517–519).

We hope these revisions adequately address the reviewer’s comments and improve the clarity and contextual depth of the manuscript.

------

It is commendable that the Limitations section acknowledges the flaws in the study, demonstrating an awareness of the study’s constraints.

Response:

We sincerely thank the reviewer for the positive feedback on the Limitations section. We greatly appreciate the recognition of our efforts to acknowledge and transparently discuss the constraints of this study.

------

While gene validation was performed using qPCR, the rationale as to the selection of two genes compared to all identified is not addressed. Were others analysed but not validated? There is no figure 2C. The number of animals included in this validation is also not specified. It is also unclear whether the same samples were used or if different ones were analyzed. These issues should be explicitly addressed in the Limitations section.

Response:

We thank the reviewer for these valuable comments. The figure numbering has been corrected accordingly (there is no Figure 2C in the second-round revised version). Regarding the RT-qPCR validation, we have clarified in the Materials and Methods section that two representative DEGs were selected solely to validate the RNA-seq results (Lines 166–169). The same RNA samples used for RNA sequencing were employed for qPCR, with n = 3 per condition, consistent with the RNA-seq analysis (Lines 169–170). As the purpose of this experiment was to confirm the consistency of expression trends rather than to provide an independent replication, we believe that this clarification in the Methods section appropriately addresses the reviewer’s concern, and therefore it was not added to the Limitations section.

------

The rationale for each statistical approach should be clearly articulated. It is important to explain why each test was chosen and why it is appropriate for such small sample sizes. Without this justification, the robustness of the statistical conclusions remains uncertain.

Response:

We appreciate the reviewer’s insightful comment. We agree that clearly articulating the rationale for each statistical method enhances confidence in the analytical robustness. Accordingly, we have revised the Materials and Methods section to include concise explanations of the statistical approaches employed.

Specifically, we now state that edgeR was chosen because it uses an empirical Bayes framework to stabilize dispersion estimates across genes, making it particularly suitable for RNA-seq data with limited biological replicates (Lines 144–146). We also added that Welch’s t-test was applied for RT-qPCR analysis to account for potential heterogeneity of variance between groups, providing more reliable inference with small sample sizes (Lines 176–178). These additions clarify that the statistical methods were selected with careful consideration of both the small sample size and the characteristics of data, thereby ensuring an appropriate and robust analytical framework.

------

To Reviewer #5

Major Points:

1. As other reviewers may have noted, the interpretation of the results is insufficient. The Results section, in particular, merely describes the findings without any interpretation, making it unclear what conclusions can be drawn from each experiment. The authors should interpret their findings within the Results section or significantly expand the Discussion to connect their data to biological meaning.

Response:

We agree with the reviewer that the Results section should primarily present the findings, whereas the Discussion section should provide their interpretation. We have maintained this standard IMRaD structure. However, we appreciate the reviewer's observation that the biological implications of each analysis were not immediately clear. To enhance readability and better guide the reader, we have revised the Results section to include concise summative statements at the end of key subsections: DEG analysis and RT-qPCR (Lines 284–287), ORA of bilateral DEGs in the environmental comparison (Lines 353–356) and GSEA in the environmental comparison (Lines 450–454). These statements succinctly summarize the direct outcomes of each analysis without introducing additional interpretation, which is reserved for the Discussion. We believe that these revisions make the presentation of the results substantially clearer, as suggested.

------

2. The authors conclude that there are significant hemispheric differences based on their results. However, it is generally expected that some differences, including experimental noise, will emerge in any omics analysis. This reviewer thinks that a more appropriate conclusion from these data would be that there are few, if any, meaningful differences between the hemispheres. The authors must define the criteria under which they would conclude a lack of hemispheric difference and provide stronger evidence that the observed differences are biologically significant, not just experimental noise or statistical artifacts.

Response:

We sincerely thank the reviewer for this critical and insightful comment, which addresses the core interpretation of our findings. We fully understand the reviewer’s skepticism: given that the direct left–right comparison yielded 0–1 DEGs, why do we conclude there is a significant hemispheric difference?

Our central claim does not concern baseline expression differences, but rather a qualitative divergence in the transcriptomic response (or “reactivity”) to environmental stimuli. The reviewer is correct that this “functional” asymmetry, derived from a limited sample size (n = 3), could potentially reflect a statistical artifact. However, we contend that the observed difference is biologically meaningful for the following reasons:

- Coherent Biological Themes: The observed differences are not random noise. Both ORA and GSEA revealed distinct, biologically coherent themes for each hemisphere—left: synaptic function, memory, and metabolism; right: RNA metabolism and transcriptional regulation. It is highly unlikely that random variation would consistently generate such distinct, functionally cohesive pathway signatures.

- Integration with External Evidence: As also suggested by other comments (e.g., Reviewer #1, Comment 4), we have explicitly linked these hemisphere-specific molecular themes to existing functional and anatomical literature. In the Discussion section (Lines 512–521), we now explain how the “metabolic and structural” theme observed in the left CA1 aligns with the concept of synaptic ultrastructure remodeling. Furthermore, we cited Shinohara et al. (2013) to propose that our observed molecular asymmetry may serve as a potential substrate for their reported functional asymmetry. This convergence with independent findings strengthens the argument that our results reflect genuine biological asymmetry rather than experimental noise.

To clarify this key point, we have also added sentences to the Discussion section (Lines 465–472) explicitly acknowledging the high baseline similarity (0–1 DEGs) while justifying our interpretation of a qualitative hemispheric asymmetry in environmental reactivity, grounded in coherent and reproducible biological themes rather than statistical artifacts. Importantly, we also emphasize that this interpretation does not constitute a definitive conclusion but rather a hypothesis that warrants validation in future large-scale studies with greater statistical power in the Limitations section (Lines 585–596).

Finally, in response to the reviewer’s question on criteria, we would have concluded that there was no hemispheric difference if both ORA and GSEA for the left and right (ISO vs. ENR) comparisons had revealed substantial overlap in enriched functional pathways. The fact that these analyses instead identified distinct and non-overlapping biological processes forms the basis for our conclusion. We believe that our revised explanation and explicit acknowledgment of the need for further validation appropriately address the reviewer’s concern.

------

3. It is inappropriate to assess hemispheric differences based solely on whether a gene passes a specific DEG threshold in one hemisphere but not the other. A gene identified as a DEG in one hemisphere might show a similar trend (i.e., similar direction and magnitude of change) in the opposite hemisphere, even if it does not meet the arbitrary p-value or fold-change cutoff. To substantiate the claim that the changes are truly hemisphere-specific, the authors should, for 150 genes, present bar graphs displaying the expression levels and corresponding p-values for both hemispheres side-by-side. This would provide a more transparent and convincing comparison.

Response:

We fully agree with the reviewer’s fundamental statistical point: assessing hemispheric differences solely by determining whether a gene passes an arbitrary DEG threshold in one hemisphere but not the other is not a robust approach. However, we would like to clarify that this is not the basis of our study’s conclusion. Our interpretation of hemispheric difference is not derived from such a “gene-by-gene” comparison.

Our argument is instead based on higher-level in silico functional analyses (ORA and GSEA). Specifically, the complete set of 189 DEGs identified in the left CA1 is enriched for “synaptic function and metabolism,” whereas the 94 DEGs in the right CA1 are associated with “transcriptional and RNA regulation” processes. The significant difference we emphasize lies in these biologically coherent pathway signatures rather than in individual gene-level contrasts.

To directly address the reviewer’s concern that genes might show similar trends across hemispheres, we performed a correlation analysis using the log2FC values for all genes between the left and right CA1 regions (S4 Fig). This analysis yielded a Pearson coefficient of r =0.587, confirming a moderate overall directional similarity in gene expression changes across hemispheres.

We believe this result actually strengthens our main argument: despite this substantial correlation in overall expression trends, the subset of the most significantly regulated genes on each side diverged into functionally distinct biological themes (synaptic vs. regulatory). This pattern is unlikely to result from a simple thresholding artifact.

Because our interpretation is drawn from pathway-level integration, and the correlation analysis already provides a comprehensive overview of interhemispheric similarity (S4 Fig), we do not believe that adding bar graphs for 150 individual genes would further clarify our central message.

------

4. This paper used an “Isolation” condition as a control compared to an “Enriched environment.” Since isolation eliminates social communication, which is critical for normal mouse development, it is questionable whether this condition serves as a proper neutral baseline. A group housing condition should have been included as a control group. This would allow for a clearer determination of how transcription becomes altered by enriched environment.

Response:

We thank the reviewer for raising this important point regarding the experimental design. We would like to clarify a crucial aspect of our study’s aim and rationale, as there may be some misunderstanding of our experimental objective.

The reviewer is correct that the “Isolation” (ISO) condition is not a physiologically neutral baseline. However, our study was not intended to use ISO as a “neutral control.” Rather, the objective was to directly compare the transcriptomic effects of two distinct and well-defined environmental conditions: social isolation (ISO) vs. sensorimotor enrichment (ENR). This specific comparison (ISO vs. ENR) was intentionally chosen because it reproduces the exact experimental paradigm used in the foundational functional literature on hemispheric lateralization, upon which our study builds.

Our goal was to identify the molecular substrates that may underlie the functional lateralization (e.g., in hippocampal gamma oscillations, as reported by Shinohara et al., 2013) observed under these same environmental manipulations. Therefore, we believe that our experimental design—which directly parallels and extends prior work in this field—is appropriate and valid for addressing our research question.

------

Minor Points:

1. Although the authors mention the duration of housing in either the isolated or enriched environment in the Discussion, this information is essential for reproducibility and should be clearly stated in the Materials and Methods section.

Response:

We thank the reviewer for this comment. The duration of housing under the isolated or enriched environment is already described in the Materials and Methods section (Lines 113–114: “Hippocampal slices were prepared... 4 weeks after rearing under the respective environmental conditions”).

------

2. The volcano plots should be annotated with gene names instead of, or in addition to, gene IDs to improve readability and

---

## [Editor Report · Decision Letter 2]

19 Nov 2025

Housing environment bilaterally alters transcriptomic profile in the rat hippocampal CA1 region

PONE-D-25-08336R2

Dear Dr. TAJIMA,

We’re pleased to inform you that your manuscript has been judged scientifically suitable for publication and will be formally accepted for publication once it meets all outstanding technical requirements.

Kind regards,

Stephen D. Ginsberg, Ph.D.

Section Editor

PLOS ONE

---

## [Editor Report · Acceptance letter]

PONE-D-25-08336R2

PLOS ONE

Dear Dr. Tajima,

I'm pleased to inform you that your manuscript has been deemed suitable for publication in PLOS ONE. Congratulations! Your manuscript is now being handed over to our production team.

Kind regards,

on behalf of

Dr. Stephen D. Ginsberg

Section Editor

PLOS ONE